# Mammal responses to global changes in human activity vary by trophic group and landscape

Wildlife must adapt to human presence to survive in the Anthropocene, so it is critical to understand species responses to humans in different contexts. We used camera trapping as a lens to view mammal responses to changes in human activity during the COVID-19 pandemic. Across 163 species sampled in 102 projects around the world, changes in the amount and timing of animal activity varied widely. Under higher human activity, mammals were less active in undeveloped areas but unexpectedly more active in developed areas while exhibiting greater nocturnality. Carnivores were most sensitive, showing the strongest decreases in activity and greatest increases in nocturnality. Wildlife managers must consider how habituation and uneven sensitivity across species may cause fundamental differences in human–wildlife interactions along gradients of human influence.

With the global human population size now past 8 billion and the associated human footprint covering much of the Earth's surface[1], survival of wild animals in the Anthropocene requires that they adapt to physical changes to the landscape and to increasing human presence. Animals often perceive humans as threats and subsequently adjust behaviours to avoid people in space or time[2]. Conversely, some animals are attracted to people to obtain resource subsidies or protection from predators[3,4]. These contrasting responses to humans shape the prospects for human–wildlife coexistence, with consequences for the capacity of human-influenced ecosystems to support robust animal populations and communities.

Variation in animal responses to human activity can be driven by intrinsic factors such as species' ecological and life-history traits (Table 1)[5]. For instance, small-bodied generalist species may be more tolerant of human presence, as they can be less conspicuous than larger species and more capable of shifting resource use within their broader niches than are specialists[6]. Wide-ranging, large-bodied carnivores face considerable risk of mortality from humans[7] and so may exhibit more risk-averse responses to human activity. Animal responses may also be heavily influenced by the type of human activity (for example, hunting versus hiking[8]) and by extrinsic factors such as landscape context. Animals may be warier of people in open or human-modified environments relative to areas with abundant vegetation cover or minimal human landscape modification[9]. Conversely, animals in heavily modified landscapes

could habituate to human presence and thus be less likely to respond to changes in human activity. Our ability to resolve such hypotheses about the interacting influences of species traits and landscape characteristics has been limited by the focus of previous studies on few species and contexts, with indirect measures of human activity and weaker correlative inferences. Ultimately, anticipating and managing impacts to wild animals requires stronger inferences from experimental manipulations of human activity and concurrent monitoring of people and animals across a range of species and environmental contexts.

Government policies during the early months of the COVID-19 pandemic (henceforth, pandemic) resulted in widespread changes to human activity that provided a quasi-experimental opportunity to study short-term behavioural responses of wild animals[10]. Early observations of animal responses to this 'anthropause'[11] relied on qualitative or opportunistic sightings prone to bias (for example, contributed by volunteers[12]), or focused on small spatial scales and few species, reporting a mix of positive and negative responses that make it difficult to reach more general conclusions[13]. Furthermore, measures of human activity have typically been coarse and indirect[14], yet changes to human activity during the pandemic appeared highly variable at the fine scales that affect animal behaviour (Fig. 1). For example, some natural areas experienced increases in human visitation while others were closed to visitors[15] and the strength of government restrictions changed over time[14]. It is thus important for studies using

✉e-mail: cole.burton@ubc.ca

**Table 1 | Predictor variables hypothesized to explain variation in species responses to higher human activity, with greater reductions in amount of activity or increases in nocturnality predicted for more sensitive species (further details in Supplementary Information)**

| Class | Variable | Prediction | Range |
|---|---|---|---|
| Species trait | Body mass | Large-bodied species will be more sensitive | Small (1–20 kg; $n=101$); large (20–4,600 kg; $n=62$) |
| Species trait | Trophic level | Higher trophic levels will be more sensitive | Carnivore ($n=59$), omnivore ($n=27$), herbivore ($n=77$) |
| Species trait | Diet breadth | Specialists with narrower diet will be more sensitive | 1–4 diet categories |
| Species trait | Habitat breadth | Specialists with narrower habitat preference will be more sensitive | 1–9 habitat categories |
| Species trait | Diel activity | Diurnal species will be most sensitive, cathemeral species intermediate and nocturnal species least sensitive | Diurnal ($n=13$), cathemeral ($n=91$), nocturnal ($n=59$) |
| Species trait | Hunting status | Hunted species (within projects) will be more sensitive to increased human activity than their non-hunted counterparts | Yes ($n=486$), no ($n=491$) (total=977 project–species) |
| Species trait | Relative brain size | Small-brained species will be more sensitive | 0.006–5.3 kg |
| Habitat structure | Openness | Animals will be more sensitive in open habitat types relative to closed habitats | Open ($n=31$), closed ($n=71$) |
| Land-use disturbance | Human modification index | Animals will be more sensitive in landscapes with more human modification | 0.005–0.834 |
| Magnitude of human change | Global stringency index | Animals will show stronger responses where lockdowns were more stringent | 38.9–96.0 stringency units |
| Magnitude of human change | Mean change in human detections (at camera traps) | Animals will show stronger responses where change in human activity greater | 1–100-fold changes |

For continuous variables we show the range (minimum–maximum); for categorical variables we show the sample size for each level, which sum to 163 species for species-level variables or 102 projects for project-level variables (unless otherwise stated). Body mass and trophic level were combined in a new variable 'trophic group'.

the pandemic as an unplanned experiment to have localized information on human activity that matches their animal data and to tackle context-dependency by using robust, standardized methods across several species and landscapes.

The widespread use of camera traps to survey terrestrial mammals[16] provides a unique opportunity to take advantage of the pandemic experiment and improve our understanding of animal responses to changes in human activity. Thousands of cameras are deployed around the world[17], providing standardized animal sampling while simultaneously quantifying local human activity[15,18]. We harnessed this opportunity to examine relationships between detections of people and mammals across gradients in land use and habitat type—spanning 102 survey sites (projects) in 21 countries (predominantly in Europe and North America) with 5,400 camera-trap locations sampling for 311,208 camera-days before and during the pandemic (Fig. 1; Methods). Some sites experienced a decrease in human activity during the pandemic, consistent with the notion of an anthropause, while there was an increase or no change at others. We focused our analysis on those sites with some change in human activity (either increase or decrease) and standardized our comparisons to be between periods of relatively lower to higher human activity (either across years or within 2020; Fig. 1; Methods) to mimic the general trend of increasing human presence in the Anthropocene. We examined site-level changes in animal detection rates and nocturnality across populations of 163 mammal species (body mass ≥ 1 kg; range 1–65 populations per species; Supplementary Table 1) as measures of the relative amount and timing of animal activity (Methods). We then used meta-analytic mixed-effects models to quantify the extent to which variation in animal responses across sites was explained by species traits, landscape modification and other site characteristics and the magnitude of change in human activity (Table 1; Methods).

## Results and discussion
Our camera-trap measures of human activity varied widely under COVID-19 lockdowns (occurring between March 2020 and January 2021), from 100-fold decreases to 10-fold increases within sites between comparison periods (Fig. 1 and Supplementary Fig. 1). These changes were not predicted by coarser measures of human activity based on the stringency of lockdowns (Supplementary Fig. 1), highlighting the complementary value of finer-scaled monitoring of human activity.

### Changes in amount of animal activity
Animals did not show consistent, negative responses to greater human activity; instead, responses were highly variable among species and sites (Figs. 2 and 3). Across 1,065 estimated responses (one per species per project, that is, population), changes in animal detection rates (reflecting the intensity of habitat use; Methods) varied from 139-fold increases to 36-fold decreases, with a near-zero mean change overall (−0.04, 95% confidence interval (CI) = −0.11–0.03; Fig. 2b). Trophic group (combining body mass and trophic level) was the strongest predictor of changes in animal activity in response to increasing human use, with large herbivores showing the largest increases in activity and carnivores showing the strongest decreases (Fig. 2c, Supplementary Table 2 and Supplementary Fig. 3). This is consistent with carnivore avoidance of higher mortality risk from encounters with people[7] and with increased herbivore activity due to either more frequent disturbance by people or attraction to human activity driven by reduced risk of predation (human shield hypothesis[3]).

Animal activity in more developed areas (that is, higher human modification index (HMI) measured at the site level; Table 1) generally increased (+25%) with higher levels of human activity, while animals in less-developed areas decreased their activity (−6%) when human activity was higher (Fig. 2c; coefficient = 0.077; 95% CI = −0.001–0.156). This contrast highlights an important interaction between human modification of a landscape and human activity therein—between human footprint and footfalls—which we posit could be the result of two factors. First, local extirpations of sensitive species (species 'filtering'[19]) would result in only human-tolerant species persisting in developed areas—for example, sensitive wolverine (*Gulo gulo*) were absent from sites with intermediate to high human modification. Second, species found across the gradient, such as mule deer (*Odocoileus hemionus*), could become habituated to benign human presence in more developed landscapes and therefore be less fearful of human activity than their conspecifics in less-developed areas[20]. Notably, this

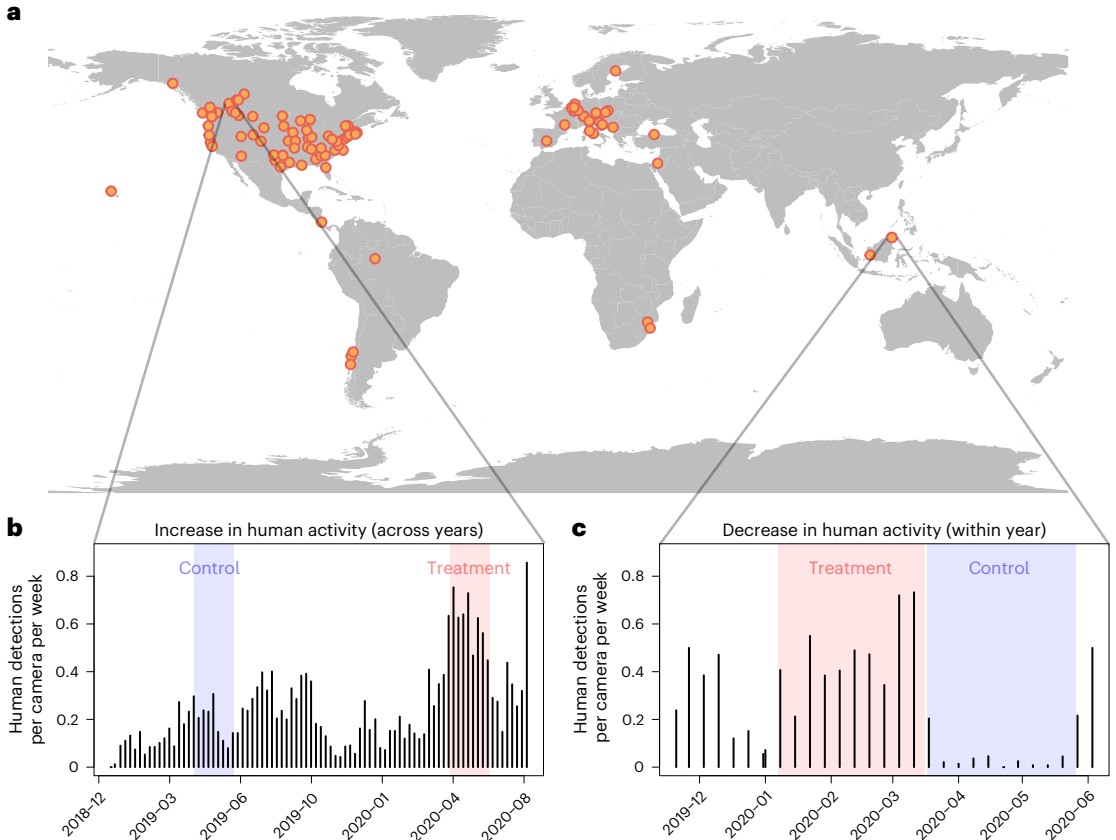

**Fig. 1 | Camera-trap sampling of contrasts between periods of higher versus lower human activity. a**, Location of camera-trap projects included in the analysis (n = 102). **b,c**, Examples for two projects: Edmonton, Canada (**b**) and Danum Valley, Malaysia (**c**) showing time series of human detections for the two types of comparisons used to assess the effects of higher human activity on animals. **b**, A between-year comparison with increased human activity during the COVID-19 pandemic (treatment, red shading) relative to the same time period the year before (control, blue shading). **c**, A within-year comparison with decreased human activity during the pandemic (control, blue shading) relative to the prepandemic period (treatment, red shading).

relationship with landscape modification varied predictably across trophic groups (Fig. 2d and Supplementary Table 3). Small and large carnivores, small herbivores and small omnivores increased their activity with higher human activity in developed areas (increasing by an average of 54%), while the response was much weaker for large herbivores and in fact opposite for large omnivores, which decreased activity when human activity increased in more modified landscapes (50% decrease; Fig. 2d). This negative response was common across all of the frequently detected large omnivores—wild boar (*Sus scrofa*), American black bear (*Ursus americanus*) and brown bear (*Ursus arctos*)—and could be driven by their attraction to anthropogenic food resources (for example garbage and fruit trees) that may be less risky to access when human activity is reduced[21].

Animal detections were also more likely to decline with higher human activity in more open habitat types such as grasslands or deserts, relative to closed habitats such as forests (Fig. 2c; coefficient = −0.172; 95% CI = −0.3428 to −0.0018). This is consistent with predictions under the landscape of fear framework that suggest that animal perceptions of risk are influenced by availability of cover[22]. Contrary to our expectations, we did not find strong evidence that the magnitude of change in human activity (measured by camera traps or the stringency index; Table 1) affected animal responses or that hunted populations changed their amount of activity more than non-hunted ones (Supplementary Tables 2, 4 and 5). We also did not find strong support for the hypothesis that species with relatively larger brains—as an index of behavioural plasticity[23]—would show more pronounced responses to changes in human activity (Supplementary Table 5).

## Changes in timing of animal activity

Whether or not animals change their intensity of use of an area, they could shift their timing of activity to minimize overlap with increasing human activity (Fig. 3a)[24]. We measured changes in animal nocturnality (proportion of night time detections) across 499 populations (Methods) and found considerable variation in animal responses to increasing human activity (though generally less than for amount of activity): from fivefold increases in nocturnality to sixfold decreases (mean change in proportion of nocturnal detections = 0.008; 95% CI = −0.02–0.04; Fig. 3b). The strongest predictor of changes in nocturnality was the degree of landscape modification (HMI): in more developed areas, animals tended to become more nocturnal as human activity increased (19.3% increase in nocturnality; Fig. 3c, coefficient = 0.047; 95% CI = 0.026–0.069; Supplementary Table 6). This is consistent with previous evidence of increasing wildlife nocturnality in the face of growing human impacts[24] and highlights the importance of the temporal refuge provided by night time cover for human–wildlife coexistence in increasingly human-dominated environments[25].

Paralleling our findings about changes in the amount of animal activity, trophic group was also an important predictor of changes in nocturnality, with large carnivores becoming notably more nocturnal than other groups (+5.3%; Fig. 3c and Supplementary Table 6). Again, we found support for an interaction between human modification and trophic group: most groups had stronger increases in nocturnality along the disturbance gradient as human activity increased (mean +22.6%), whereas the increases in nocturnality for large carnivores did not vary with land-use disturbance (Fig. 3d and Supplementary Table 7). This finding could reflect greater

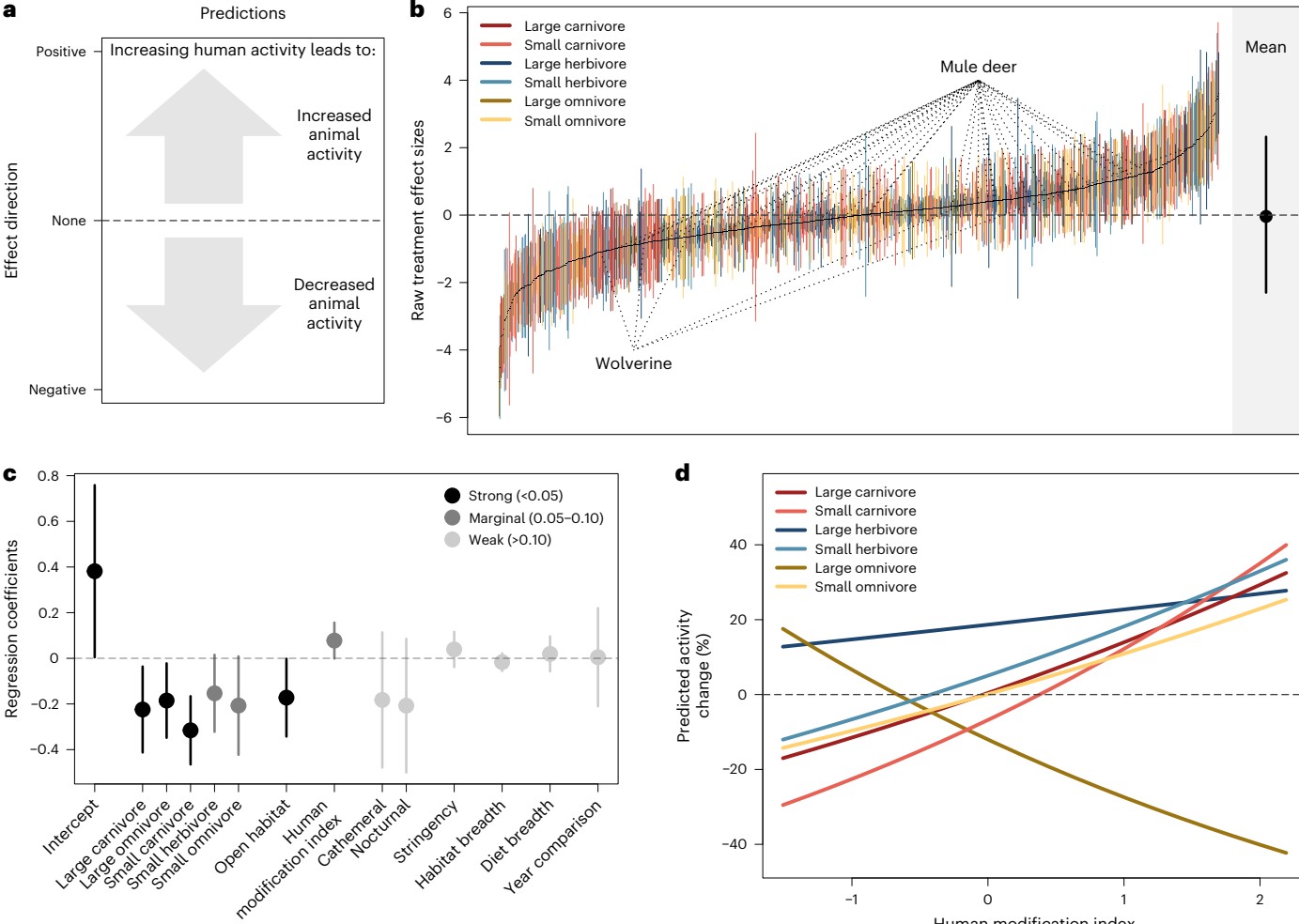

**Fig. 2 | Changes in the amount of animal activity in response to increasing human activity. a**, Interpretation of effects. **b**, Estimated effect sizes (black points) and variances (coloured lines) for all populations included in the analysis ($n = 1,065$ project–species combinations from 102 independent projects; two example species highlighted) with the global mean (and 95% quantiles) plotted in black to the right. **c**, Estimated model coefficients (points) and 95% CIs (lines; $n = 1,065$ project–species combinations from 102 independent projects) for additive factors (with complete data; Methods) hypothesized to influence changes in the amount of animal activity when human activity is higher, where: intercept is diurnal, large herbivore in closed habitat type with a seasonal comparison and all other effects are contrasts. **d**, Model predictions for the interaction between trophic group and HMI.

sensitivity of large carnivores to the increased risk of conflict associated with more human presence[26], such that they shift timing of activity to minimize overlap regardless of landscape context. Other groups increased night time activity only in landscapes with higher risk of human encounters (that is, more modification), which may in turn enable the increases in amount of activity observed for many of these species (Fig. 2d).

Unlike for the amount of activity, changes in the timing of animal activity were mediated by the hunting status of species in an area, whereby hunted animals showed stronger increases in nocturnal behaviour at higher levels of landscape modification (+26.6%) relative to their non-hunted counterparts (+13.5%; Fig. 3e and Supplementary Table 8). We did not find strong evidence that relative brain size was associated with shifts in animal nocturnality, nor that the magnitude of change in the amount of human activity explained variation in animal responses (Fig. 3c and Supplementary Tables 6 and 9). We did find an effect of our comparison type such that, on average, comparisons between years showed larger shifts in nocturnality than within-year comparisons (Fig. 3c and Supplementary Table 6), underscoring the importance of temporal matching to minimize influence of other factors such as seasonal changes in activity patterns.

## Implications for human–wildlife coexistence

Contrary to popular narratives of animals roaming more widely while people sheltered in place during early stages of the COVID-19 pandemic, our results reveal tremendous variation and complexity in animal responses to dynamic changes in human activity. Using a unique synthesis of simultaneous camera-trap sampling of people and hundreds of mammal species around the world, combined with a powerful before–after quasi-experimental design, we quantified how animals change their behaviours under higher levels of human activity across gradients of human footprint. As the human population continues to grow, the persistence of wild animals will depend on their responses to increasing human presence in both highly and moderately modified landscapes. It may thus be encouraging that many animal populations did not show dramatic changes in the amount or timing of their activity under conditions of higher human activity. Indeed, mean changes across all populations assessed were close to zero, suggesting that there was no global systematic shift in animal activity during the pandemic, consistent with other recent observations of highly variable animal responses[13,27]. Nevertheless, we saw stronger responses to human activity for certain species and contexts and these patterns can help us better understand and mitigate negative impacts of people on wildlife communities.

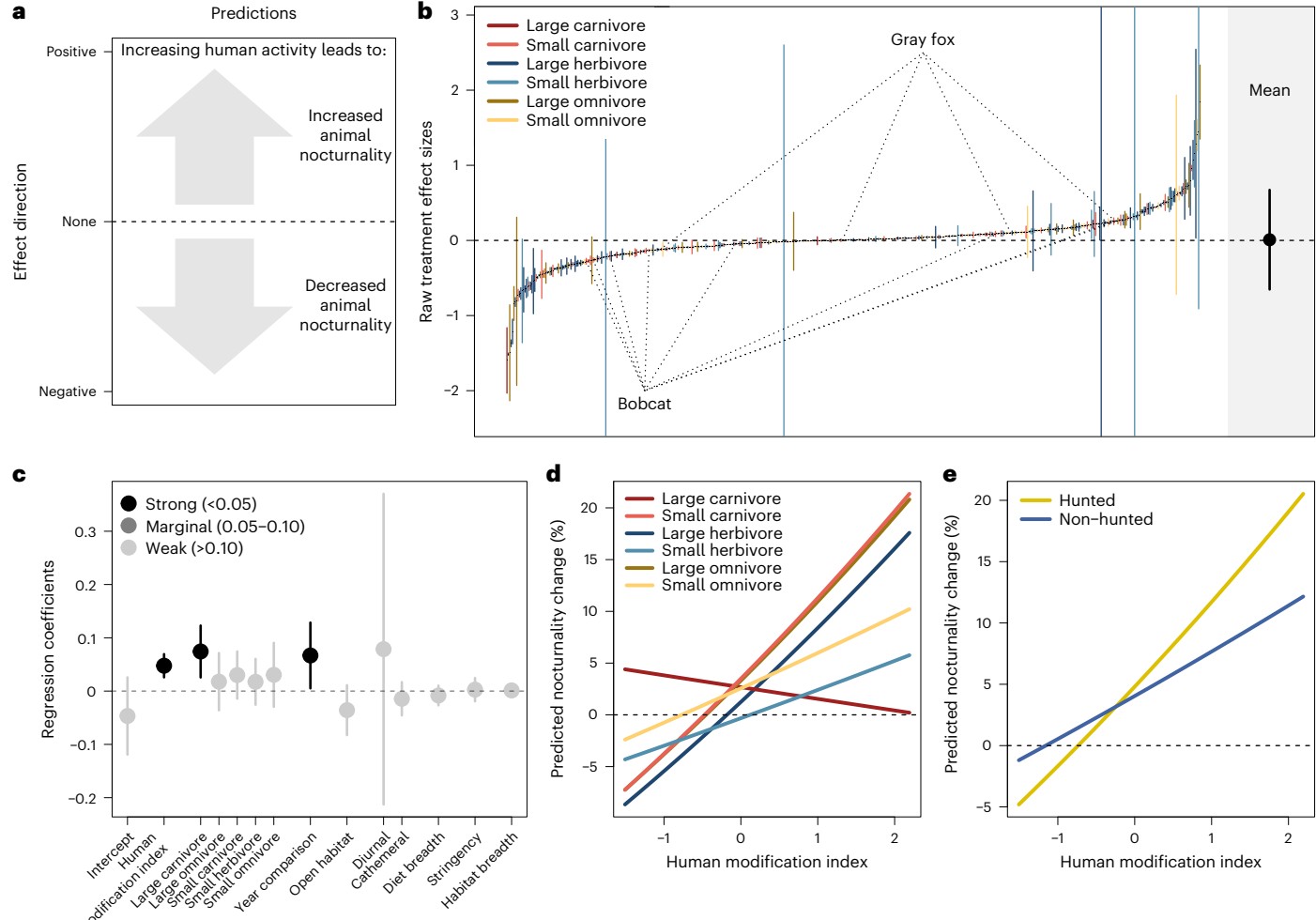

**Fig. 3 | Changes in animal nocturnality in response to increasing human activity. a**, Interpretation of effects. **b**, Estimated effect sizes (black points) and variances (coloured lines) for all populations included in the analysis ($n = 499$ project–species combinations from 100 independent projects; two example species highlighted) with the global mean (with 95% quantiles) plotted in black to the right. **c**, Estimated model coefficients (points) and 95% CIs (lines; $n = 499$ project–species combinations from 100 independent projects) for additive factors (with complete data; Methods) hypothesized to influence changes in animal nocturnality when human activity is higher, where: intercept is nocturnal, large herbivore in closed habitat type with a seasonal comparison and all other effects are contrasts. **d**, Model predictions for interaction between trophic group and human modification index. **e**, Model predictions for interaction between hunting and HMI.

One striking pattern is that animal responses to human activity varied with the degree of human landscape modification. Our results imply that risk tolerance and associated behaviours vary between wildlife in more- versus less-developed contexts. As human activity increased, many species in more modified landscapes surprisingly had higher overall activity, although this activity was more nocturnal, suggesting that animals persisting in these developed environments may be attracted to anthropogenic resource subsidies but still seek ways to minimize encounters with people through partitioning time[28]. Wildlife managers in such modified environments should anticipate some animal habituation and manage the timing of human activity to protect night time refuges that promote human–wildlife coexistence—particularly for hunted species that showed the strongest shifts toward nocturnality. On the other hand, regulating the amount of human activity may be more important in less-developed landscapes where we detected the greatest declines in animal activity with increasing human activity. Such remote landscapes are often spatial refuges for sensitive species that may be filtered out as human modification increases; yet these areas face increasing demands from popular pursuits, such as outdoor recreation and nature-based tourism[18], and may also be more difficult to protect from illegal hunting, encroachment or resource extraction[29].

The sensitivity of species to human footprint and footfalls varied by trophic group and body size, as did the interplay of space and time in behavioural responses. Both large and small carnivore species were among the more sensitive to changes in human activity, generally reducing their activity levels and exhibiting more nocturnality with higher human activity. This motivates a continued emphasis on carnivore behaviour and management as a key challenge for human–wildlife coexistence, given the threatened status of many carnivores, the risk of negative outcomes of human–carnivore encounters and the ecological importance of carnivores as strongly interacting species[7,30]. Avoidance of people by carnivores could be beneficial if it reduces human–carnivore conflict[25,28] but it could also lead to different types of conflict if it results in lower predation rates on herbivores near people, as seen in overbrowsing by habituated deer[4]. Indeed, large herbivores showed the strongest increases in activity with higher human activity in our study, consistent with habituation and increased risk of conflict. Large omnivores, such as bear and boar, were unique in both spatially and temporally avoiding higher human activity in more developed environments, underscoring that management efforts to regulate human activity and create spatial or temporal refuges may lead to outcomes that differ by species and setting. Managers must pay particular attention to the

prospect that such differential responses can alter species interactions and cause knock-on effects with broader consequences for ecosystem functions and services[31,32].

Our study highlights the value of learning from unplanned 'experiments' caused by rapid changes in human activity[33] and other extreme events (for example, ref. 34). These insights are enabled by sampling methods, such as camera trapping, that facilitate standardized, continuous monitoring of diverse animal assemblages and humans across varied landscape contexts. While many studies of the anthropause focused on wildlife observations by volunteers in more accessible urban environments (for example, ref. 35), our results emphasize that animal responses to changes in human activity differ between more- and less-developed landscapes. This context-dependency should be a focus of further research, including expanded assessment of contexts and species under-represented in our sample, such as those in tropical regions subjected to different pressures during the pandemic[36]. Many geographic and taxonomic gaps in global biodiversity monitoring remain and must be filled by cost-effective networks that gather reliable evidence across several scales; standardized camera-trap programmes and infrastructure are helping to do so[37,38]. As the cumulative effects of the human enterprise put pressure on ecosystems worldwide[39], bending the curve of biodiversity loss will require context-specific knowledge on ecological responses to human actions that can guide locally appropriate and globally effective conservation solutions.

## Methods

### Data collection

We issued a call in September 2020 to camera-trap researchers around the world for contributions of camera-trap data from before and during the onset of the COVID-19 pandemic and associated restrictions on human activity[10,11]. This initial call included a social media post (Twitter, now X) and targeted emails to 143 researchers in 37 countries. We requested datasets that adhered to global camera-trap metadata standards (Wildlife Insights[38]) and received submissions from 146 projects. Submitted data were summarized using a standardized script and evaluated according to the following key criteria: (1) most or all camera-trap stations were deployed in the same area of interest (hereafter site) before and during COVID-19-related restrictions; (2) a minimum of seven unique camera-trap deployment locations (stations) were sampled; (3) a minimum sampling effort of at least 7 days per camera period (see below); and (4) trends in human detections were recorded from camera-trap data (that is, detections of humans) or human activity for a given sampling area was available from other sources (for example, lockdown dates and local knowledge).

We only included detections of wild mammal species ≥1 kg (mean species body mass in kg obtained from ref. 40; we excluded domestic animals, which represented only 6% of overall detections and were associated with humans) and humans (excluding research personnel servicing cameras). Our full dataset for the next step of analysis included 112 projects sampling across 5,653 cameras for 329,535 camera-days (see below for data included in specific models). The mean number of camera locations per project was 42 (range 6–300) and mean camera-days per project was 2,945 (range 348–27,986). Camera locations were considered independent within projects, as no paired cameras were included (see Supplementary Table 10 for more details on camera deployments and spacing).

### Experimental design

For each project, we first reviewed site-level trends in independent detection events of humans (using a standardized 30 min interval: that is, a detection was considered independent if >30 min from previous detection at the same camera station) to identify whether there were changes in human activity associated with COVID-19 restrictions in 2020. We sought to identify two comparable sampling periods that differed in human activity but were otherwise similar (for example,

in camera locations and sampling effort) and thus could be used as a quasi-experimental comparison to assess wildlife responses to the change in human activity. We initially anticipated that human activity would be reduced during COVID-19 lockdowns (that is, the anthropause[11]) but observed a wide variety of patterns of human detections across datasets, including decreases, increases and no change in human detections between sampling before and during COVID-19 (Supplementary Fig. 1). Since our primary interest was in evaluating wildlife responses to changes in human activity and in general we anticipate increases in human activity during the Anthropocene, we standardized our treatments to represent increases in human activity. In other words, we defined a 'control' period as one with lower human activity and a 'treatment' period as one with higher human activity, regardless of which occurred before or during the COVID-19 pandemic (Fig. 1).

We identified start and end dates for each period on the basis of clear changes in human detections (determined from visual inspection of daily detections; Fig. 1). For some projects, dates corresponded to known dates of local COVID-19 lockdowns or changes in study design (for example, dates of camera placement or removal). We prioritized comparison between years when data were collected in similar periods in years before 2020 (n = 95 projects). If multiyear data were not available, we selected comparison periods before and after the onset of lockdowns around March 2020 (with specific dates chosen according to local lockdown conditions; n = 17). If there were several potential treatment periods, we prioritized periods on the basis of the following ordered criteria: (1) the fewest seasonal or ecological confounds; (2) the most similar study design; (3) the greatest sampling effort; and (4) the most recent time period. Of the 95 projects for which we made comparisons between 2020 and a previous year, we used 2019 for 88 projects, 2018 for 6 and 2017 for 1.

In cases where there was no noticeable difference in human detections between candidate periods, or there were insufficient human detections from camera traps, we used other data or local knowledge of changes in human activity (for example, lockdown dates and visitor use data) from co-authors responsible for the particular project. Of the 112 projects included in our initial analyses, 15 used this expert opinion to determine changes in human activity. After completing our initial categorization of comparison periods, we shared details with all data contributors for review and adjustment, if necessary, based on expert knowledge of a given study area. Contributors were asked whether our delineation of sampling periods as being high versus low in human activity corresponded with their knowledge of the study system. We also asked them to consider whether other sources of environmental variation (for example, fire, drought, seasonal or interannual variation) or sampling design could confound the attribution of changes in wildlife detections to changes in human activity. After this evaluation and review, we retained 102 project datasets that had a detectable change in human activity between a treatment and control period for subsequent statistical modelling. These projects spanned 21 countries, mostly in North America and Europe but with some representation from South America, Africa and Southeast Asia (Fig. 1 and Supplementary Table 10).

Our paired treatment–control design makes several assumptions. For instance, we assumed that either: (1) changes in human activity occurred in the same direction throughout the entire study area within the treatment period; (2) the direction of the average effect was more important than variation in direction across camera sites; (3) variation in human activity within a study area was lower than differences in human activity between the treatment (higher activity) and control (lower activity) periods. By standardizing our treatment to be the period of higher human activity, we also assumed that the temporal direction of change did not affect animal responses.

### Data analysis

We compared two response variables between treatment and control periods to assess wildlife responses to changes in human activity: the

amount of animal activity and the timing of animal activity (described below). We used a two-stage approach in which we first estimated the direction and magnitude of change in these responses between periods for each species and then used a meta-analytical approach to evaluate the degree to which a set of candidate predictor variables explained variation in estimated responses. All data manipulation and analysis were done using R statistical software (v.4.1.3; ref. [41]).

**Amount of animal activity.** To evaluate changes in the amount of animal activity, we quantified detection rates for each mammal species (and humans) at each camera for the treatment and control periods of each project. Specifically, we calculated the number of independent detections for a given species and camera station using a standardized 30 min interval (that is, detection was considered independent if >30 min from previous detection of the same species at the same camera station), while controlling for variation in sampling effort (log of camera-days included as an offset in models). We assumed that this detection rate (sometimes termed relative abundance index[16]) measured the relative intensity of habitat use by a species at a camera station, which reflects both the local abundance of the species (number of individuals in sampled area) and the movement patterns of individuals.

To quantify the magnitude of change in the amount of animal activity, we first ran single-species models to estimate changes in detection rates for species and humans between the comparison periods for each project. The response variable was the count of independent detection events, modelled as negative binomial, with an offset for active camera-days. Treatment was included as a fixed effect and a random intercept was included for camera station where the same camera locations were sampled in both periods (no random effect was included if a project used different camera locations between periods). All models were implemented using the glmmTMB package[42]. These models produced a regression coefficient (effect size) for each project–species population (humans and animals) representing the estimated magnitude of change in the amount of activity between the control period and the treatment period (and its corresponding sampling variance).

**Timing of animal activity.** To assess changes in timing of animal activity, we first classified each independent detection of a given species within a given project as 'day' or 'night'. We used the lutz package to convert all local times to UTC[43]. We calculated the angle of the sun at the time of the first image in each detection using the sunAngle function in the oce package[44], based on the UTC time and latitude and longitude of the camera deployment location. Negative sun angles corresponded to 'night' (between sunset and sunrise) and positive sun angles to 'day' (between sunrise and sunset). Following ref. [24], we calculated an index of nocturnality, $N$, as the proportion of independent camera-trap detections that occurred during the night ($N$ = detections during night/ (detections during night + detections during day)) for all species which had ten or more detections in both the control and treatment periods. We then calculated the log risk ratio, RR and its corresponding sampling variance (weighted by sample size) between the treatment and control periods, pooled across all camera traps within a given study using the escalc() function within the metafor package[45]. This effect size compared the percentage of animal detections that occurred at night with high human activity ($N_h$) to night time animal activity under low human activity ($N_l$), with RR = $\ln(N_h/N_l)$). A positive RR indicated a relatively greater degree of nocturnality in response to human activity, while a negative RR indicated reduced nocturnality.

**Hypothesized explanatory variables.** We identified and calculated a set of variables that we hypothesized would affect species responses to changes in human activity. These fell into four general classes: (1) species traits, (2) habitat (that is, vegetation) structure, (3) anthropogenic landscape modification and (4) magnitude of human change

(Table 1). We did not include any covariates reflecting differences in camera-trap sampling protocols between projects, as our estimates of species responses were made within projects (that is, comparing treatment versus control periods) and thus sampling methods were internally consistent within projects (for example, camera placement and settings).

**Species traits.** We hypothesized that species with the following traits would be more sensitive to changes in human activity (that is, more vulnerable or risk averse): larger body mass[46], higher trophic level[46], narrower diet and habitat breadth[47], diurnal activity[46] and smaller relative brain size[48]. We extracted variables for each species from the COMBINE database[40], the most comprehensive archive of several mammal traits curated to date (representing 6,234 species). Given that some traits in the database were imputed, we reviewed the designations for plausibility and cross-referenced the traits with other widely used databases—specifically Elton Traits[49] and PanTHERIA[50]—and made the following corrections to the 'activity cycle' trait (diurnal, nocturnal and cathemeral): diurnal to cathemeral—*Mellivora capensis, Neofelis nebulosa, Neofelis diardi*; diurnal to nocturnal—*Meles meles;* nocturnal to diurnal—*Phacochoerus africanus;* nocturnal to cathemeral—*Ursus americanus.* To calculate relative brain size we divided log-transformed brain mass by log-transformed body mass (as in ref. [48]). We combined body mass and trophic level into a new variable 'trophic group' (consisting of small- or large-bodied categories for each of the three trophic levels, Table 1). Dietary and habitat breadth are described in ref. [40].

We further hypothesized that animals in hunted populations would be more sensitive to changes in human activity. We requested that all data contributors complete a survey indicating whether a given species was hunted within their project survey area, from which we created a binary factor representing hunting status for each population (1 = hunted; 0 = not hunted).

**Habitat structure.** Camera-trap surveys included in our analysis covered an extensive range of biogeographic areas and habitat types. We made the simplifying assumption that species responses to changes in human activity would be most influenced by the degree of openness of habitat (that is, vegetation structure) in a sampling area. More specifically, we hypothesized that areas with more open habitat types would have higher visibility and thus less security cover for animals and thus that animals in these open habitats would be more sensitive to increases in human activity than would animals in more closed habitats with greater security cover[51]. We used the Copernicus Global Land Cover dataset (100 m resolution[52]) via Google Earth Engine to extract land cover class at each camera station. We then used the percentage canopy cover of the mode class across all cameras in a given project to define if the survey occurred in primarily closed (>70% canopy cover) or open habitat types (0–70% canopy cover).

**Land cover disturbance.** We posited that animal responses to changes in human activity would differ according to the degree of anthropogenic landscape modification (that is, human footprint[1,53]). More specifically, we identified two hypotheses that could underlie variation in species responses as a function of land cover disturbance. On the one hand, our 'habituation hypothesis' predicts that animals in more disturbed landscapes may be less sensitive to changes in human activity (relative to animals in undisturbed landscapes) and thus show less of a negative response or even a positive response as they have already behaviourally adapted to tolerate co-occurrence with people[22]. On the other hand, our 'plasticity hypothesis' predicts that the ability of animals to coexist with people in disturbed landscapes may be dependent on plasticity in animal behaviour[22], such that animals in these landscapes may show more pronounced and rapid responses to changes in human activity (for example, avoidance of areas and times with greater chance of encountering people).

We initially characterized landscape disturbance using three variables accessed via Google Earth Engine: Gridded Population of the World (1 km resolution[54]), road density (m km$^{-2}$, 8 km resolution; Global Roads Inventory Project[55]) and HMI (for 2016 at 1 km resolution), which represents a cumulative measure of the proportion of a landscape modified by 13 anthropogenic stressors[53]. Point values were extracted for each camera station in each site, then the project-level medians were used in analysis. As the median values of these three variables were highly correlated across projects (Supplementary Fig. 2), we only used HMI in our subsequent models.

**Magnitude of human change.** We expected that animal responses would be more pronounced in areas that underwent greater changes in human activity and we used two measures to assess the magnitude of those changes. At a coarse scale, we used the COVID-19 stringency index[14], which characterizes the policies restricting human activities within a given geographic region at a daily time scale and has been widely used in studies of COVID-19 on human mobility and the environment (for example, ref. [13]). We used the finest-scale regional data available for each project, which was usually at the country level, with the exception of three countries with province- or state-level data (Brazil, Canada and the United States). When projects spanned several countries, provinces or states, we used the stringency index for the region in which most cameras were located. For each region, we calculated the median stringency for the treatment and control sampling periods.

At a finer scale, we used the effect size for the modelled change in camera-trap detection rates of humans across all cameras in a project (as described above under 'amount of animal activity'). Models with this variable excluded 15 projects that either did not detect humans with camera traps or the number of humans detected on cameras was not perceived by the data contributor to be an accurate reflection of change in human use for the sampled area.

**Meta-analysis models.** To understand which factors mediated the effect of increasing human use on animal activity, we ran mixed-effect meta-analytic models using the rma.mv() function of the metafor package[45] on the effect sizes and sampling variances of the two response variables described above (amount and timing of animal activity). Our unit of observation for modelling was the estimated response for each project–species combination (that is, each animal population) and we included random intercepts for project and for species nested within family, to account for repeated observations within each of those higher-level groups and for phylogenetic relatedness within families. All continuous predictor variables (Table 1) were standardized to unit variance with a mean of zero using the stdize function in the MuMIn package[56]. We tested pairwise correlations among all predictor variables and found that none were highly correlated (that is, all below a threshold of Pearson |*r*| < 0.6; Supplementary Fig. 2) and thus all were retained for modelling.

We performed our analysis in three steps for each of the two wildlife response variables. First, we fit a global model including all hypothesized predictor variables for which we had complete data (excluding hunting status, relative brain size and empirical magnitude of human change, for which we had incomplete data and thus included in analysis of subsets of data, described below). Second, we used model selection to test for plausible interactions and nonlinear effects. Third, we used model selection on subsets of the full data to compare the global and interactions models with candidate models adding three more predictor variables with incomplete data.

**Global model.** As all of our predictor variables were independent, we used a global model approach that included additive fixed effects for all predictor variables (Table 1). We interpreted the *P* value of each effect contrast to indicate statistically significant support (at *P* < 0.05 or marginal support at *P* < 0.10) for a consistent effect direction of a given predictor and we used the estimated effect size as a measure of effect

magnitude. We calculated the pseudo-$R^2$ to estimate the total variation explained by our global models. We also calculated the $I^2$ (ref. [57]) of each global model to determine the amount of heterogeneity observed between the random effect levels; consistent variation in the response terms between projects, families and species would result in higher $I^2$ values compared to the null model with no fixed effects. To aid interpretation, we present effect sizes in terms of the proportional change (%) in model-predicted responses across lowest-to-highest values for continuous predictors (for example, HMI) or between two categories of interest (for example, trophic groups).

**Model selection of plausible interactions and nonlinear terms.** To explore the possibility of context-specific effects of the predictors of wildlife responses to changes in human activity, we assessed a suite of ecologically plausible interaction and nonlinear (quadratic) terms through adding them in turn to the global model and using Akaike's Information Criterion (corrected for small sample size, AICc) to find the most parsimonious model. We assessed the following terms: (1) 'HMI * habitat_closure', to evaluate the potential for habitat structure to mediate responses to human landscape modification; (2) 'trophic_group * HMI', to evaluate the potential for different trophic groups to respond to human modification in different ways; (3) 'trophic_group * habitat_closure', to evaluate the potential for different trophic groups to respond to habitat structure in different ways; and (4) HMI$^2$, to assess nonlinear effects of wildlife responses to human modification. Models including the candidate interaction or nonlinear terms were compared to the global model without interaction terms using AICc (in the MuMIn package[56]) and were discussed above if they were within 2 AICc of the best-supported model and there was no simpler, nested model with more support.

**Model selection on subsets of data.** We had a small amount of missing information in the data available for assessing the effects of population hunting status, species relative brain size and empirical (that is, camera-trap-based) magnitude of change in human activity (91.7%, 98.8% and 86.5% of project–species had data for these variables, respectively). Therefore, we ran the same global model used for the full dataset on the subsetted data along with candidate models including each of these predictor variables and all plausible interactions of interest (as above). These additional candidate models were compared to the global model (run on the same partial dataset) using AICc and were discussed in the results if they resulted in a lower AICc value (that is, had more support than the global model, which was a simpler nested model).

### Reporting summary
Further information on research design is available in the Nature Portfolio Reporting Summary linked to this article.

## Data availability
The data used in this paper are available in Figshare, with the identifier: https://doi.org/10.6084/m9.figshare.23506536.

## Code availability
The code used to analyse the data and create the figures in this paper are available in Figshare, with the identifier: https://figshare.com/articles/software/Analysis_R_Code/23506512.

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

## Acknowledgements

We recognize the tragic consequences of the COVID-19 pandemic and would like to acknowledge all people impacted. Full acknowledgements are provided in the Supplementary Information. This synthesis project was funded by the Natural Sciences and Engineering Research Council of Canada (Canada Research Chair 950-231654 and Discovery Grant RGPIN-2018-03958 to A.C.B. and RGPIN-2022-03096 to K.M.G.) and the National Center for Ecological Analysis and Synthesis (Director's Postdoc Fellowship to K.M.G.). Additional funding sources for component subprojects are listed in the Supplementary Information.

## Author contributions

A.C.B., C. Beirne, R.K., K.M.G., C. Sun and A. Granados conceived this work. A.C.B., C. Beirne, R.K., K.M.G., A. Granados, C. Sun and F.C. were responsible for data curation. C. Beirne and K.M.G. conducted the formal analysis. A.C.B., R.K. and K.M.G. acquired funding. A.C.B., C. Beirne, K.M.G., C. Sun, A. Granados, M.L.A., J.M.A., G.C.A., F.S.Á.C., Z.A., C.A.-D., C.A., S.A.-A., G.B., A.B.-M., D.B., E.B., E.L.B., C. Baruzzi, S.M.B., N. Beenaerts, J. Belmaker, O.B., B.B., T.B., D.A.B., N. Bogdanović, A.B., M.B., L.B., J.F.B., J. Brooke, J.W.B., F.C., B.S.C., J. Carvalho, J. Casaer, R. Černe, R. Chen, E.C., M.C., C. Cincotta, D.Ć., T.D.C., J. Compton, C. Coon, M.V.C., A.P.C., S.D.F., A.K.D., M. Davis, K.D., V.D.W., E.D., T.A.D., J.D., M. Duľa, S.E.-F., C.E., A.E., J.F.-L., J. Favreau, M.F., P.F., F.F., C.F., L.F., J.T.F., M.C.F.-R., E.A.F., U.F., J.Fl., J.M.F., A.F., B. Franzetti, S. Frey, S. Fritts, Š. Frýbová, B. Furnas, B.G., H.M.G., D.G.G., A.J.G., T.G., M.E.G., D.M.G., M.G., A. Green, R.H., R.(B.)H., S. Hammerich, C. Hanekom, C. Hansen, S. Hasstedt, M. Hebblewhite, M. Heurich, T.R.H., T.H., D.J., P.A.J., K.J.J., A.J., M.J., M.C.K., M.J.K., M.T.K., S.K.-S., M. Krofel, A.K., K.M.K., D.P.J.K., E.K.K., J.K., M. Kutal, D.J.R.L., S.L., M. Lashley, R. Lathrop, T.E.L.J., C.L., D.B.L., A.L., M. Linnell, J. Loch, R. Long, R.C.L., J. Louvrier, M.S.L., P.M., S.M., B.M., G.K.H.M., A.J.M., D.M., Z.M., T.M., W.J.M., M.M., C.M., J.J.M., C.M.M.-M., D.M.-A., K.M., C. Nagy, R.N., I.N., C. Nelson, B.O., M.T.O., V.O., C.O., F.O., P.P., K.P., L.P., C.E.P., M. Pendergast, F.F.P., R.P., X.P.-O., M. Price, M. Procko, M.D.P., E.E.R., N.R., S.R., K.R., M.R., R.R., R.R.-H., D.R., E.G.R., A.R., C. Rota, F.R., H.R., C. Rutz, M. Salvatori, D.S., C.M.S., J. Scherger, J. Schipper, D.G.S., Ç.H.Ş., P.S., J. Sevin, H.S., C. Shier, E.A.S.-R., M. Sindicic, L.K.S., A.S., T.S., C.C.S.C., J. Stenglein, P.A.S., K.M.S., M. Stevens, C. Stevenson, B.T., I.T., R.T.T., J.T., T.U., J.-P.V., D.V., S.L.W., J. Weber, K.C.B.W., L.S.W., C.A.W., J. Whittington, I.W., M.W., J. Williamson, C.C.W., T.W., H.U.W., Y.Z., A.Z. and R.K. carried out the investigations. A.C.B., R.K. and F.C. were responsible for project administration. A.C.B., C. Beirne, R.K., K.M.G., C. Sun and A. Granados wrote the original draft manuscript. A.C.B., C. Beirne, K.M.G., C. Sun, A. Granados, M. Lashley, J.M.A., G.C.A., F.S.Á.C., Z.A., C.A.-D., C.A., S.A.-A., G.B., A.B.-M., D.B., E.B., E.L.B., C. Baruzzi, S.M.B., N. Beenaerts, J. Belmaker, O.B., B.B., T.B., D.A.B., N. Bogdanović, A.B., M.B., L.B., J.F.B., J. Brooke, J.W.B., F.C., B.S.C., J. Carvalho, J. Casaer, R. Černe, R. Chen, E.C., M.C., C. Cincotta, D.Ć., T.D.C., J. Compton, C. Coon, M.V.C., A.P.C., S.D.F., A.K.D., M. Davis, K.D., V.D.W., E.D, T.A.D., J.D., M. Duľa, S.E.-F., C.E., A.E., J.F.-L., J. Favreau, M.F., P.F., F.F., C.F., L.F., J.T.F., M.C.F.-R., E.A.F., U.F., J.Fl., J.M.F., A.F., B. Franzetti, S. Frey, S. Fritts, Š. Frýbová, B. Furnas, B.G., H.M.G., D.G.G., A.J.G., T.G., M.E.G., D.M.G., M.G., A. Green, R.H., R.(B.)H., S. Hammerich, C. Hanekom, C. Hansen, S. Hasstedt, M. Hebblewhite, M. Heurich, T.R.H., T.H., D.J., P.A.J., K.J.J., A.J., M.J., M.C.K., M.J.K., M.T.K., S.K.-S., M. Krofel, A.K., K.M.K., D.P.J.K., E.K.K., J.K., M. Kutal, D.J.R.L., S.L., M. Lashley, R. Lathrop, T.E.L.J., C.L., D.B.L., A.L., M. Linnell, J. Loch, R. Long, R.C.L., J. Louvrier, M.S.L., P.M., S.M., B.M., G.K.H.M., A.J.M., D.M., Z.M., T.M., W.J.M., M.M., C.M., J.J.M., C.M.M.-M., D.M.-A., K.M., C. Nagy, R.N., I.N., C. Nelson, B.O., M.T.O., V.O., C.O., F.O., P.P., K.P., L.P., C.E.P., M. Pendergast, F.F.P., R.P., X.P.-O., M. Price, M. Procko, M.D.P., E.E.R., N.R., S.R., K.R., M.R., R.R., R.R.-H., D.R., E.G.R., A.R., C. Rota, F.R., H.R., C. Rutz, M. Salvatori, D.S., C.M.S., J. Scherger, J. Schipper, D.G.S., Ç.H.Ş., P.S., J. Sevin, H.S., C. Shier, E.A.S.-R., M. Sindicic, L.K.S., A.S., T.S., C.C.S.C., J. Stenglein, P.A.S., K.M.S., M. Stevens, C. Stevenson, B.T., I.T., R.T.T., J.T., T.U., J.-P.V., D.V., S.L.W., J. Weber, K.C.B.W., L.S.W., C.A.W., J. Whittington, I.W., M.W., J. Williamson, C.C.W., T.W., H.U.W., Y.Z., A.Z. and R.K. were involved in reviewing and editing the final manuscript.

## Competing interests

The authors declare no competing interests.

## Additional information

**Correspondence and requests for materials** should be addressed to A. Cole Burton.

A. Cole Burton [1,2,161] ✉, Christopher Beirne[1,161], Kaitlyn M. Gaynor[2,3,4], Catherine Sun[1], Alys Granados[1], Maximilian L. Allen [5], Jesse M. Alston [6], Guilherme C. Alvarenga [7], Francisco Samuel Álvarez Calderón [8], Zachary Amir [9], Christine Anhalt-Depies [10], Cara Appel [11], Stephanny Arroyo-Arce [12], Guy Balme[13], Avi Bar-Massada [14], Daniele Barcelos [7], Evan Barr[15], Erika L. Barthelmess [16], Carolina Baruzzi [17], Sayantani M. Basak [18], Natalie Beenaerts [19], Jonathan Belmaker [20], Olgirda Belova[21], Branko Bezarević [22], Tori Bird [23], Daniel A. Bogan[24], Neda Bogdanović [25], Andy Boyce [26], Mark Boyce [27], LaRoy Brandt [28], Jedediah F. Brodie [29,30], Jarred Brooke[31], Jakub W. Bubnicki[32], Francesca Cagnacci [33,34], Benjamin Scott Carr [35], João Carvalho [36], Jim Casaer[37], Rok Černe[38], Ron Chen [39], Emily Chow[40], Marcin Churski [32], Connor Cincotta[41], Duško Ćirović[25], T. D. Coates[42], Justin Compton [43], Courtney Coon[44], Michael V. Cove [45], Anthony P. Crupi [46], Simone Dal Farra[33], Andrea K. Darracq[15], Miranda Davis [47], Kimberly Dawe[48], Valerie De Waele[49], Esther Descalzo[50], Tom A. Diserens[32,51], Jakub Drimaj [52], Martin Duľa [52,53], Susan Ellis-Felege[54], Caroline Ellison [55], Alper Ertürk [56], Jean Fantle-Lepczyk[57], Jorie Favreau[41], Mitch Fennell [1], Pablo Ferreras [50], Francesco Ferretti[34,58], Christian Fiderer[59,60], Laura Finnegan [61], Jason T. Fisher [62], M. Caitlin Fisher-Reid [63], Elizabeth A. Flaherty [31], Urša Fležar[38,64], Jiří Flousek[65], Jennifer M. Foca [27], Adam Ford[66], Barbara Franzetti [67], Sandra Frey[62], Sarah Fritts[68], Šárka Frýbová[69], Brett Furnas[70], Brian Gerber [71], Hayley M. Geyle [72], Diego G. Giménez[73], Anthony J. Giordano[73], Tomislav Gomercic[74], Matthew E. Gompper [75], Diogo Maia Gräbin[7], Morgan Gray [76], Austin Green[77], Robert Hagen[78,79], Robert (Bob) Hagen[80], Steven Hammerich[76], Catharine Hanekom [81], Christopher Hansen[82], Steven Hasstedt [83], Mark Hebblewhite [29], Marco Heurich [59,60,84], Tim R. Hofmeester [85], Tru Hubbard[86], David Jachowski[87], Patrick A. Jansen [88,89], Kodi Jo Jaspers[90], Alex Jensen[87], Mark Jordan [91], Mariane C. Kaizer [92], Marcella J. Kelly [93], Michel T. Kohl[35], Stephanie Kramer-Schadt [79,94], Miha Krofel [64], Andrea Krug[95], Kellie M. Kuhn [83], Dries P. J. Kuijper[32], Erin K. Kuprewicz [47], Josip Kusak[74], Miroslav Kutal [52,53], Diana J. R. Lafferty [86], Summer LaRose[96], Marcus Lashley[97], Richard Lathrop[98], Thomas E. Lee Jr[99], Christopher Lepczyk [57], Damon B. Lesmeister[100], Alain Licoppe [49], Marco Linnell[100], Jan Loch[101], Robert Long[90], Robert C. Lonsinger [102], Julie Louvrier [79], Matthew Scott Luskin [9], Paula MacKay[90], Sean Maher [103], Benoît Manet[49], Gareth K. H. Mann[13], Andrew J. Marshall [104], David Mason [97], Zara McDonald[44], Tracy McKay[61], William J. McShea[26], Matt Mechler[105], Claude Miaud [106], Joshua J. Millspaugh[82], Claudio M. Monteza-Moreno[107], Dario Moreira-Arce [108], Kayleigh Mullen[23], Christopher Nagy[109], Robin Naidoo [110], Itai Namir[20], Carrie Nelson[111], Brian O'Neill[112], M. Teague O'Mara [113], Valentina Oberosler [114], Christian Osorio [115], Federico Ossi [33,34], Pablo Palencia [116,117], Kimberly Pearson[118], Luca Pedrotti[119], Charles E. Pekins[120], Mary Pendergast[121], Fernando F. Pinho[7], Radim Plhal[52], Xochilt Pocasangre-Orellana[8], Melissa Price[122], Michael Procko [1], Mike D. Proctor [123], Emiliano Esterci Ramalho [7], Nathan Ranc[33,124], Slaven Reljic [74], Katie Remine[90], Michael Rentz[125], Ronald Revord[96], Rafael Reyna-Hurtado[126], Derek Risch [122], Euan G. Ritchie[127], Andrea Romero [112], Christopher Rota [128], Francesco Rovero[114,129], Helen Rowe [130,131], Christian Rutz [132], Marco Salvatori [114,129], Derek Sandow[133], Christopher M. Schalk [134], Jenna Scherger[66], Jan Schipper [135], Daniel G. Scognamillo[136], Çağan H. Şekercioğlu[77,137], Paola Semenzato[138], Jennifer Sevin[139], Hila Shamon[26], Catherine Shier [140], Eduardo A. Silva-Rodríguez [141], Magda Sindicic[74], Lucy K. Smyth[13,142], Anil Soyumert [56], Tiffany Sprague[130], Colleen Cassady St. Clair [27], Jennifer Stenglein [10], Philip A. Stephens [143], Kinga Magdalena Stępniak [144], Michael Stevens[145], Cassondra Stevenson[27], Bálint Ternyik[143,146], Ian Thomson[12], Rita T. Torres [36], Joan Tremblay[47], Tomas Urrutia[115], Jean-Pierre Vacher[106], Darcy Visscher [147], Stephen L. Webb[148], Julian Weber[149], Katherine C. B. Weiss [135], Laura S. Whipple[150], Christopher A. Whittier [151], Jesse Whittington [152], Izabela Wierzbowska [18], Martin Wikelski [107,153], Jacque Williamson [154], Christopher C. Wilmers[155], Todd Windle[156], Heiko U. Wittmer[157], Yuri Zharikov[158], Adam Zorn[159] & Roland Kays [45,160]

[1]Department of Forest Resources Management, University of British Columbia, Vancouver, British Columbia, Canada. [2]Biodiversity Research Centre, University of British Columbia, Vancouver, British Columbia, Canada. [3]Departments of Zoology and Botany, University of British Columbia, Vancouver, British Columbia, Canada. [4]National Center for Ecological Analysis and Synthesis, Santa Barbara, CA, USA. [5]Illinois Natural History Survey, Prairie Research Institute, University of Illinois, Champaign, IL, USA. [6]School of Natural Resources and the Environment, University of Arizona, Tucson, AZ, USA. [7]Instituto de Desenvolvimento Sustentável Mamirauá, Tefé, Brazil. [8]Fundación Naturaleza El Salvador, San Salvador, El Salvador. [9]School of Biological Sciences, University of Queensland, Brisbane, Queensland, Australia. [10]Wisconsin Department of Natural Resources, Madison, WI, USA. [11]College of Agricultural Sciences, Oregon State University, Corvallis, OR, USA. [12]Coastal Jaguar Conservation, Heredia, Costa Rica. [13]Panthera, New York, NY, USA. [14]Department of Biology and Environment, University of Haifa at Oranim, Kiryat Tivon, Israel. [15]Watershed Studies Institute, Murray State University, Murray, KY, USA. [16]St. Lawrence University, Canton, NY, USA. [17]School of Forest, Fisheries and Geomatics Sciences, University of Florida, Gainesville, FL, USA. [18]Institute of Environmental Sciences, Faculty of Biology, Jagiellonian University, Kraków, Poland. [19]Centre for Environmental Sciences, Hasselt University, Hasselt, Belgium. [20]School of Zoology, Faculty of Life Sciences, Tel Aviv University, Tel Aviv, Israel. [21]Institute of Forestry, Lithuanian Research Centre for Agriculture and Forestry, Kėdainių, Lithuania. [22]National Park Tara, Mokra Gora, Serbia. [23]Hogle Zoo, Salt Lake City, UT, USA. [24]Siena College, Loudonville, NY, USA. [25]Faculty of Biology, University of Belgrade, Belgrade, Serbia. [26]Smithsonian's National Zoo and Conservation Biology Institute, Washington, DC, USA. [27]Department of Biological Sciences, University of Alberta, Edmonton, Alberta, Canada. [28]Lincoln Memorial University, Harrogate, TN, USA. [29]Division of Biological Sciences & Wildlife Biology Program, University of Montana, Missoula, MT, USA. [30]Institute of Biodiversity and Environmental Conservation,

Universiti Malaysia Sarawak, Kota Samarahan, Malaysia. [31]Purdue University, West Lafayette, IN, USA. [32]Mammal Research Institute, Polish Academy of Sciences, Białowieża, Poland. [33]Animal Ecology Unit, Research and Innovation Centre, Fondazione Edmund Mach, Trento, Italy. [34]National Biodiversity Future Center (NBFC), Palermo, Italy. [35]Warnell School of Forestry and Natural Resources, University of Georgia, Athens, GA, USA. [36]Department of Biology and Centre for Environmental and Marine Studies, University of Aveiro, Aveiro, Portugal. [37]Research Institute for Nature and Forest, Brussels, Belgium. [38]Slovenia Forest Service, Ljubljana, Slovenia. [39]Hamaarag, Steinhardt Museum of Natural History, Tel Aviv University, Tel Aviv, Israel. [40]British Columbia Ministry of Forests, Cranbrook, British Columbia, Canada. [41]Paul Smith's College, Paul Smiths, NY, USA. [42]Royal Botanic Gardens Victoria, Melbourne, Victoria, Australia. [43]Springfield College, Springfield, MA, USA. [44]Felidae Conservation Fund, Mill Valley, CA, USA. [45]North Carolina Museum of Natural Sciences, Raleigh, NC, USA. [46]Alaska Department of Fish and Game, Juneau, AK, USA. [47]University of Connecticut, Storrs, CT, USA. [48]Quest University Canada, Squamish, British Columbia, Canada. [49]Service Public of Wallonia, Gembloux, Belgium. [50]Instituto de Investigación en Recursos Cinegéticos, Ciudad Real, Spain. [51]Faculty of Biology, University of Warsaw, Warsaw, Poland. [52]Faculty of Forestry and Wood Technology, Mendel University in Brno, Brno, Czech Republic. [53]Friends of the Earth Czech Republic, Carnivore Conservation Programme, Olomouc, Czech Republic. [54]University of North Dakota, Grand Forks, ND, USA. [55]Texas Parks and Wildlife Department, Austin, TX, USA. [56]Hunting and Wildlife Program, Kastamonu University, Kastamonu, Turkey. [57]College of Forestry, Wildlife and Environment, Auburn University, Auburn, AL, USA. [58]Department of Life Sciences, University of Siena, Siena, Italy. [59]Bavarian Forest National Park, Grafenau, Germany. [60]University of Freiburg, Breisgau, Germany. [61]fRI Research, Hinton, Alberta, Canada. [62]University of Victoria, Victoria, British Columbia, Canada. [63]Bridgewater State University, Bridgewater, MA, USA. [64]Biotechnical Faculty, University of Ljubljana, Ljubljana, Slovenia. [65]Krkonoše Mountains National Park, Vrchlabí, Czech Republic. [66]Department of Biology, University of British Columbia, Kelowna, British Columbia, Canada. [67]Italian Institute for Environmental Protection and Research, Rome, Italy. [68]Texas State University, San Marcos, TX, USA. [69]Department of Botany and Zoology, Faculty of Science, Masaryk University, Brno, Czech Republic. [70]California Department of Fish and Wildlife, Sacramento, CA, USA. [71]University of Rhode Island, Kingstown, RI, USA. [72]Research Institute for the Environment and Livelihoods, Charles Darwin University, Darwin, Northern Territory, Australia. [73]Society for the Preservation of Endangered Carnivores and their International Ecological Study (S.P.E.C.I.E.S.), Ventura, CA, USA. [74]Faculty of Veterinary Medicine, University of Zagreb, Zagreb, Croatia. [75]New Mexico State University, Las Cruces, NM, USA. [76]Pepperwood, Santa Rosa, CA, USA. [77]University of Utah, Salt Lake City, UT, USA. [78]Agricultural Center for Cattle, Grassland, Dairy, Game and Fisheries of Baden-Württemberg, Aulendorf, Germany. [79]Leibniz Institute for Zoo and Wildlife Research, Berlin, Germany. [80]University of Kansas, Lawrence, KS, USA. [81]Ezemvelo KZN Wildlife, Pietermaritzburg, South Africa. [82]University of Montana, Missoula, MT, USA. [83]US Air Force Academy, Colorado Springs, CO, USA. [84]Inland Norway University, Hamar, Norway. [85]Department of Wildlife, Fish and Environmental Studies, Swedish University of Agricultural Sciences, Umeå, Sweden. [86]Northern Michigan University, Marquette, MI, USA. [87]Clemson University, Clemson, SC, USA. [88]Smithsonian Tropical Research Institute, Balboa, Republic of Panama. [89]Department of Environmental Sciences, Wageningen University and Research, Wageningen, the Netherlands. [90]Woodland Park Zoo, Seattle, WA, USA. [91]Seattle University, Seattle, WA, USA. [92]National Institute of the Atlantic Forest, Santa Teresa, Brazil. [93]Virginia Tech, Blacksburg, VA, USA. [94]Institute of Ecology, Technische Universität Berlin, Berlin, Germany. [95]BUND Niedersachsen, Hanover, Germany. [96]University of Missouri, Columbia, MO, USA. [97]Department of Wildlife Ecology and Conservation, University of Florida, Gainesville, FL, USA. [98]Rutgers University, New Brunswick, NJ, USA. [99]Abilene Christian University, Abilene, TX, USA. [100]United States Department of Agriculture Forest Service, Pacific Northwest Research Station, Corvallis, OR, USA. [101]Scientific Laboratory of Gorce National Park, Niedźwiedź, Poland. [102]South Dakota State University, Brookings, SD, USA. [103]Missouri State University, Springfield, MO, USA. [104]University of Michigan, Ann Arbor, MI, USA. [105]City of Issaquah, Issaquah, WA, USA. [106]CEFE, Univ Montpellier, CNRS, EPHE-PSL University, IRD, Montpellier, France. [107]Department of Migration, Max Planck Institute of Animal Behaviour, Konstanz, Germany. [108]Universidad de Santiago de Chile (USACH) and Institute of Ecology and Biodiversity (IEB), Santiago, Chile. [109]Mianus River Gorge, Bedford, MA, USA. [110]World Wildlife Fund—USA, Washington, DC, USA. [111]Effigy Mounds National Monument, Harper's Ferry, WV, USA. [112]University of Wisconsin-Whitewater, Whitewater, WI, USA. [113]Southeastern Louisiana University, Hammond, LA, USA. [114]Museo delle Scienze (MUSE), Trento, Italy. [115]Carnivoros Australes, Talca, Chile. [116]University of Castilla-La Mancha Instituto de Investigación en Recursos Cinegéticos, Ciudad Real, Spain. [117]Department of Veterinary Sciences, University of Torino, Turin, Italy. [118]Parks Canada—Waterton Lakes National Park, Waterton Park, Alberta, Canada. [119]Stelvio National Park, Bormio, Italy. [120]United States Army, Fort Hood, TX, USA. [121]Sageland Collaborative, Salt Lake City, UT, USA. [122]University of Hawai'i at Manoa, Honolulu, HI, USA. [123]Noble Research Institute, LLC, Ardmore, OK, USA. [124]Université de Toulouse, INRAE, CEFS, Castanet-Tolosan, France. [125]Iowa State University, Ames, IA, USA. [126]El Colegio de la Frontera Sur, Campeche, Mexico. [127]Centre for Integrative Ecology, School of Life and Environmental Sciences, Deakin University, Melbourne, Victoria, Australia. [128]West Virginia University, Morgantown, WV, USA. [129]Department of Biology, University of Florence, Florence, Italy. [130]McDowell Sonoran Conservancy, Scottsdale, AZ, USA. [131]Northern Arizona University, Flagstaff, AZ, USA. [132]Centre for Biological Diversity, School of Biology, University of St Andrews, St Andrews, UK. [133]Northern and Yorke Landscape Board, Clare, South Australia, Australia. [134]United States Department of Agriculture Forest Service, Southern Research Station, Nacogdoches, TX, USA. [135]Arizona State University, West, Glendale, AZ, USA. [136]Stephen F Austin State University, Nacogdoches, TX, USA. [137]Koç University, Istanbul, Turkey. [138]Research, Ecology and Environment Dimension (D.R.E.A.M.), Pistoia, Italy. [139]University of Richmond, Richmond, VA, USA. [140]Planning and Environmental Services, City of Edmonton, Edmonton, Alberta, Canada. [141]Instituto de Conservación, Biodiversidad y Territorio & Programa Austral Patagonia, Facultad de Ciencias Forestales y Recursos Naturales, Universidad Austral de Chile, Valdivia, Chile. [142]iCWild, Department of Biological Sciences, University of Cape Town, Cape Town, South Africa. [143]Conservation Ecology Group, Department of Biosciences, Durham University, Durham, UK. [144]Department of Ecology, Institute of Functional Biology and Ecology, Faculty of Biology, University of Warsaw, Warsaw, Poland. [145]Parks Victoria, Melbourne, Victoria, Australia. [146]United Nations Environment Programme World Conservation Monitoring Centre (UNEP-WCMC), Cambridge, UK. [147]The King's University, Edmonton, Alberta, Canada. [148]Natural Resources Institute and Department of Rangeland, Wildlife and Fisheries Management, Texas A&M University, College Station, TX, USA. [149]Oeko-Log Freilandforschung, Friedrichswalde, Germany. [150]University of Illinois, Urbana, IL, USA. [151]Tufts University, Grafton, MA, USA. [152]Parks Canada, Banff, Alberta, Canada. [153]Department of Biology, University of Konstanz, Konstanz, Germany. [154]Wildlife Habitat Council, Silver Spring, MD, USA. [155]Environmental Studies Department, University of California Santa Cruz, Santa Cruz, CA, USA. [156]Parks Canada, Alberni-Clayoquot, British Columbia, Canada. [157]Victoria University of Wellington, Wellington, New Zealand. [158]Parks Canada, Ucluelet, British Columbia, Canada. [159]University of Mount Union, Alliance, OH, USA. [160]North Carolina State University, Raleigh, NC, USA. [161]These authors contributed equally: A. Cole Burton, Christopher Beirne. ✉e-mail: cole.burton@ubc.ca

# Reporting Summary

## Statistics

For all statistical analyses, confirm that the following items are present in the figure legend, table legend, main text, or Methods section.

| n/a | Confirmed | |
|---|---|---|
| ☐ | ☒ | The exact sample size (*n*) for each experimental group/condition, given as a discrete number and unit of measurement |
| ☐ | ☒ | A statement on whether measurements were taken from distinct samples or whether the same sample was measured repeatedly |
| ☐ | ☒ | The statistical test(s) used AND whether they are one- or two-sided<br>*Only common tests should be described solely by name; describe more complex techniques in the Methods section.* |
| ☐ | ☒ | A description of all covariates tested |
| ☐ | ☒ | A description of any assumptions or corrections, such as tests of normality and adjustment for multiple comparisons |
| ☐ | ☒ | A full description of the statistical parameters including central tendency (e.g. means) or other basic estimates (e.g. regression coefficient) AND variation (e.g. standard deviation) or associated estimates of uncertainty (e.g. confidence intervals) |
| ☐ | ☒ | For null hypothesis testing, the test statistic (e.g. *F*, *t*, *r*) with confidence intervals, effect sizes, degrees of freedom and *P* value noted<br>*Give P values as exact values whenever suitable.* |
| ☒ | ☐ | For Bayesian analysis, information on the choice of priors and Markov chain Monte Carlo settings |
| ☐ | ☒ | For hierarchical and complex designs, identification of the appropriate level for tests and full reporting of outcomes |
| ☐ | ☒ | Estimates of effect sizes (e.g. Cohen's *d*, Pearson's *r*), indicating how they were calculated |

*Our web collection on statistics for biologists contains articles on many of the points above.*

## Software and code

Policy information about availability of computer code

| Data collection | No software was used to collect data |
|---|---|
| Data analysis | R Statistical Software, version 4.1.3. Code for data analysis are available on FigShare (link provided in Code Availability statement in paper). |

For manuscripts utilizing custom algorithms or software that are central to the research but not yet described in published literature, software must be made available to editors and reviewers. We strongly encourage code deposition in a community repository (e.g. GitHub). See the Nature Portfolio guidelines for submitting code & software for further information.

## Data

Policy information about availability of data

All manuscripts must include a data availability statement. This statement should provide the following information, where applicable:

- Accession codes, unique identifiers, or web links for publicly available datasets
- A description of any restrictions on data availability
- For clinical datasets or third party data, please ensure that the statement adheres to our policy

All data and code have been made available on FigShare with the links provided in the Data Availability and Code Availability statements in the paper.

## Human research participants

Policy information about studies involving human research participants and Sex and Gender in Research.

| | |
|---|---|
| Reporting on sex and gender | NA (our study did not involve human research participants) |
| Population characteristics | NA |
| Recruitment | NA |
| Ethics oversight | NA |

Note that full information on the approval of the study protocol must also be provided in the manuscript.

# Field-specific reporting

Please select the one below that is the best fit for your research. If you are not sure, read the appropriate sections before making your selection.

☐ Life sciences          ☐ Behavioural & social sciences          ☒ Ecological, evolutionary & environmental sciences

For a reference copy of the document with all sections, see nature.com/documents/nr-reporting-summary-flat.pdf

# Ecological, evolutionary & environmental sciences study design

All studies must disclose on these points even when the disclosure is negative.

| | |
|---|---|
| Study description | Comparison of amount and timing of animal activity between paired treatment (higher human activity) and control (lower human activity) time periods using detections from 5400 camera traps for 163 species across 102 survey areas. Unit of comparison was the population (species-project), with 1065 for amount of activity and 499 for timing of activity. |
| Research sample | Sample of terrestrial mammals detected at motion-triggered camera traps across 102 survey areas around the world. Surveys were identified opportunistically as those with active camera trap sampling before and during the COVID-19 lockdowns in 2020. Within surveys, cameras were deployed randomly or systematically to detect medium- and large-bodied terrestrial mammals (≥ 1 kg), including humans. |
| Sampling strategy | We included surveys for which: most or all camera trap stations were deployed in the same area of interest before and during COVID-19-related restrictions; a minimum of 7 unique camera trap deployment locations were sampled; a minimum sampling effort of at least 7 days per camera period; and trends in human detections were recorded from camera trap data or human activity was available from other sources. |
| Data collection | Mammals photographed by camera traps were identified from images by researchers from each project. The date and time of each detection was recorded, as was the location of each camera trap. |
| Timing and spatial scale | The spatial scale includes the entire world as we considered camera trap surveys from anywhere that met our criteria (listed above under sampling strategy). The timing of sampling varied by project, and across all projects spanned from 2017-2020, with most sampling between 2019-2020. |
| Data exclusions | We received data submissions from 146 projects, of which 112 met our sampling criteria (described above). We analyzed data from 102 projects, excluding 10 projects that did now show any change in human activity (i.e., no treatment effect). |
| Reproducibility | This was not a controlled experiment but rather a quasi-experiment based on changes in human activity in response to COVID-19 policies. Our samples were therefore not reproducible. |
| Randomization | Our comparisons of human and animal activity were between time periods within survey areas, thus controlling for variation between survey areas. |
| Blinding | Animals are detected by passive infrared cameras triggered by animal motion and body heat. There is no researcher bias in detecting animals. |

Did the study involve field work?     ☒ Yes     ☐ No

# Field work, collection and transport

| | |
|---|---|
| Field conditions | Our study covered 102 survey areas around the world, and compared animal detections between time periods. Field conditions thus varied substantially |
| Location | Our study covered 102 survey areas around the world. |
| Access & import/export | Camera trap sampling is non-invasive and does not involved any capture, handling or collection of animal specimens. |
| Disturbance | Camera trap sampling is non-invasive and causes minimal disturbance to animals. |

# Reporting for specific materials, systems and methods

We require information from authors about some types of materials, experimental systems and methods used in many studies. Here, indicate whether each material, system or method listed is relevant to your study. If you are not sure if a list item applies to your research, read the appropriate section before selecting a response.

## Materials & experimental systems

| n/a | Involved in the study |
|---|---|
| ☒ ☐ | Antibodies |
| ☒ ☐ | Eukaryotic cell lines |
| ☒ ☐ | Palaeontology and archaeology |
| ☐ ☒ | Animals and other organisms |
| ☒ ☐ | Clinical data |
| ☒ ☐ | Dual use research of concern |

## Methods

| n/a | Involved in the study |
|---|---|
| ☒ ☐ | ChIP-seq |
| ☒ ☐ | Flow cytometry |
| ☒ ☐ | MRI-based neuroimaging |

# Animals and other research organisms

Policy information about studies involving animals; ARRIVE guidelines recommended for reporting animal research, and Sex and Gender in Research

| | |
|---|---|
| Laboratory animals | The study did not involve laboratory animals. |
| Wild animals | No animals were captured or handled. Animals were observed through photographic records obtained by passive infrared remotely triggered cameras |
| Reporting on sex | Both sexes for all species were recorded by photographs. There is no reason to expect any sex bias in sampling. |
| Field-collected samples | Study did not involve samples collected from the field (only photographs) |
| Ethics oversight | No ethical approval is required for non-invasive photographic sampling by passive, remote infrared cameras |

Note that full information on the approval of the study protocol must also be provided in the manuscript.

