## [Peer Review File · Nature Ecology & Evolution]

Peer Review Information

Journal: Nature Ecology & Evolution

Manuscript Title: Mammal responses to global changes in human activity vary by trophic group and landscape

Corresponding author name(s): A. Cole Burton

Editorial Notes:

Reviewer Comments & Decisions:

Decision Letter, initial version:

26th April 2023

Dear Dr Burton,

Your manuscript entitled "Mammal responses to global changes in human activity vary by trophic group and landscape" has now been seen by 3 reviewers, whose comments are attached. The reviewers have raised a number of concerns which will need to be addressed before we can offer publication in Nature Ecology & Evolution. We will therefore need to see your responses to the criticisms raised and to some editorial concerns, along with a revised manuscript, before we can reach a final decision regarding publication.

In particular, Referee #1 feels that more detail is needed on how landscape variation is accounted for, and also how uncertainty among sampling sites has been accounted for. Referee #2 feels that more needs to be done to highlight the novel results in the study, and they also have methodological concerns over the incorporation of human activity, and the evaluation of sample sizes and model fits. Referee #3 also shares these concerns over the human activity data, as well as the use of the HMI. They also feel that the bias toward European/US sites needs to be more clearly discussed.

We therefore invite you to revise your manuscript taking into account all reviewer and editor comments. Please highlight all changes in the manuscript text file [OPTIONAL: in Microsoft Word format].

* If you have not done so already please begin to revise your manuscript so that it conforms to our Article format instructions at <http://www.nature.com/natecolevol/info/final-submission>. Refer also to

2any guidelines provided in this letter.

[REDACTED]

Nature Ecology & Evolution is committed to improving transparency in authorship. As part of our efforts in this direction, we are now requesting that all authors identified as 'corresponding author' on published papers create and link their Open Researcher and Contributor Identifier (ORCID) with their account on the Manuscript Tracking System (MTS), prior to acceptance. ORCID helps the scientific community achieve unambiguous attribution of all scholarly contributions. You can create and link your ORCID from the home page of the MTS by clicking on 'Modify my Springer Nature account'. For more information please visit www.springernature.com/orcid.

[REDACTED]

Reviewer expertise:

Reviewer #1: Urban ecology, movement, camera traps

Reviewer #2: Urban ecology, movement, mammals, camera traps

Reviewer #3: Wildlife monitoring, camera traps, human-wildlife conflict

Reviewers' comments:

Reviewer #1 (Remarks to the Author):

Review for:

Mammal responses to global changes in human activity vary by trophic group and landscape

NOTE: I have written this review in markdown format. I also attached a PDF if that is easier for you to read.

In this paper the authors amassed a huge collection of camera trap data and used a quasi-experimental approach to assess how mammals modified their relative activity patterns and diel behavior in response to a shift in human activity caused by COVID-19 lockdowns. Overall, this paper was an absolute pleasure to read. The writing was fantastic, the figures are solid, and the story that is told from these data is very interesting. Perhaps some of the most novel results from this analysis is the shifts in activity patterns as a function of human modification and trophic guilds. Given the scale of the data, this finding represents very solid evidence that mammal responses substantially vary depending on their trophic guild in predictable in understandable ways. As such, I think this paper would be of interest to a broad readership.

Based on my reading of the paper, I have two larger concerns.

1. Based on reading the paper (and not the methods) a reader does not especially know if the landscape variation the paper assesses is within or among different camera trap studies. Being more specific about this would really help given that it helps the reader understand the scale at which inference is being made. In fact, given that the scale does appear to be at the project level (now that I have read them), there should likely be some reference to how the results here represent larger-scale averages right? I'm thinking that even within a camera trap study there is going to be variation (e.g., along an urban gradient within a study) and this variation is not being assessed, which is totally fine for this scale of analysis. However, I can easily see readers getting confused about how to interpret the overall effects here (i.e., interpret them at the camera trap level instead of the project level) which is not ideal. Providing some nuance on this, as well some appropriate caveats about what this analysis therefore represents, would help.

2. From my reading of the methods (and the fact that code was not provided) I could not figure out what the uncertainty metric is that got created for the risk ratios for the nocturnality analysis. It seems like they exist (I discuss this a bit in the methods section at the bottom of the review) but that was not made clear in the supplemental methods section. Essentially what I want to make certain of is that there is some weighting by sample size that is occurring given that there is likely going to be some large scale variation in sample size across species and projects. Proportions generated from thousands of records should have more weight / more certainty than those generated from just a handful.

3These issues, however, seem like they can be addressed with just a small amount of revision of the text.

Great work on this, if you have any questions about the comments here feel free to reach out to me at mfidino@lpzoo.org. I have comments for each section below as well, I hope you find them useful.

- Mason Fidino

Abstract

Top-level thoughts

Only thing I think I would add into this abstract is a tiny peppering of quantitative results. For example, the abstract states mammals were less active in rural areas under higher human activity. On average, what was the magnitude of this relationship? Adding something like `mammals were, on average, X times less active in rural areas...` would be welcome (though I suppose how representative an average like this across so many species and projects is something to consider as well). Overall , great synopsis!

Introduction

Top-level thoughts

1. The second paragraph uses `animals` but in reality I think the authors are more specifically referencing `mammals`. For example, birds of larger size are actually more tolerant of human presence, not small-bodied (Neate-Clegg et al. 2023). This is in opposition to the point the authors make on lines 276-278. I'm splitting hairs here, but given that the title of the manuscript has already brought the focus down to mammals I think being more specific here would be warranted.

````

Montague H.C. Neate-Clegg, Benjamin A. Tonelli, Casey Youngflesh, Joanna X. Wu, Graham A. Montgomery, Çağan H. Şekercioğlu, Morgan W. Tingley. (2023).  
Traits shaping urban tolerance in birds differ around the world,  
Current Biology  
<https://doi.org/10.1016/j.cub.2023.03.024>.

````

Line by line comments

4Line 292-305: Great points here.

Changes in amount of animal activity

Top-level thoughts

1. At this stage I am not sure what scale these regression coefficients are, and as a result I am uncertain of the magnitude of effect when they are reported. As just one example, I am unsure how big the effect size really is for the results on lines 354 - 357. How much did animal activity increase with higher levels of human activity? Figure 2C does not help with this either. In fact, the whole paragraph on lines 354 - 375 only references direction of effect without providing any reference to how large this differences are. To me, this appears to be a really big missed opportunity based on those slopes in Fig. 2D and raw treatment effects on Fig 2.B. Certainly there are some space constraints, but it would be great to add some more quantitative statements about these results in this section (because it seems like these responses are quite large for some species).

Line by line comments

Line 346: Any uncertainty in that estimate (i.e., -0.04)?

Line 346 - 349: This sentence is a little bit confusing because the reference to the response variable (activity) happens quite far into the sentence. This makes the first part of the sentence, before the comma, a little confusing. What does it mean that trophic ground is a predictor of responses to increasing human use? A bit more clarity here would help the reader, especially because the methods are supplemental. However, I am not a personal fan of calling out something as confusing without at least attempting a suggested revision. Maybe something like this?

```

How animal activity rates changed in response to increasing human use varied by trophic group (combining body mass and trophic level), with large herbivores showing the largest increases in activity and carnivores the strongest decreases (Fig. 2C, etc. etc.).

```

Line 350: maybe use ``mortality risk`` instead of ``risk of mortality`` so you don't have two of's chained together.

Line 354-357: Is this at the project level or within a project? In other words, is this variation along a gradient of human modification within a camera trap study or is it variation in the average human modification index across different camera trap studies?

Changes in timing of animal activity

5Top-level thoughts

1. Same comment applies here as above. Some more quantification of effect size in the text would help.

Implications for human-wildlife coexistence

Top-level thoughts

1. The take-away from the paragraph on lines 453 - 467 hinges on whether this among-species average has meaning. It means very little when you focus your attention to large-carnivores, for example. I think you are providing that nuance in the last line of the paragraph, but not sure if `...certain species and contexts...` is an explicit call out to this?

Methods

NOTE: This is by line number within the supplemental material

Top-level thoughts

1. For the timing of animal activity. Is there any uncertainty associated to the log risk ratio? It would seem to me that there should be less weight for deployments / species with less data but from my reading of 174 - 187 it seems like every data point would be equally weighted in subsequent analyses. To me, it would seem like you would want to

a) Fit some type of Bayesian analysis (e.g., logistic regression) to get the estimated proportion of nighttime animal activity at high and low human activity levels. This would result in a posterior, so you would calculate your risk ratio with uncertainty.

b) Fit a frequentist logistic regression but do some bootstrapping. This would also result in a distribution of proportions.

In my opinion, the Bayesian analysis would be easier than a bootstrap, especially if you need to add a random effect. However, it does seem like there is some uncertainty that was created given that Fig 3B has 95% CI? If there is uncertainty then that should be made more clear here as I am uncertain what the sampling variance is that gets provided into the meta-analytic model.

2. The justification for not accounting for variation in sampling among projects is confusing (lines 193 - 196) given that the covariates that got calculated represent differences in sampling among projects (i.e., where cameras were placed). Do the authors instead mean variation in study design for projects (e.g., how high cameras were placed, if lure was used, etc.)?

3. When calculating independent detections it appears that this was done at the species and camera level. However, it does not appear that there was any checking to determine if cameras were independent of one another? Some studies could have had paired cameras, and so the same individual could be getting captured on two "independent cameras." I'm assuming the authors accounted for this, but adding a sentence in the methods here with respect to this would help. Likewise, it could help to report the minimum distance between cameras (average across studies, min and max).

4. A sentence on what the human modification index represents would help with the interpretation of your results. I had to google it to find out what it represented.

Line by line comments

Line 156-157: Was sampling effort a log offset term or just included in the model? I'm assuming an offset (as it should be) but that information is left out. Wait, I see it now on lines 164 (it's an offset), just hint at it here and you are golden.

NOTE: I added my name above, but the reviewer instructions also asked me to put it here.

Mason Fidino

Reviewer #2 (Remarks to the Author):

The authors here investigate the effects of human activity associated with 2020 COVID pandemic on mammal activity and nocturnality. They integrate several explanatory variables based primarily on species traits and interactions with habitat and human modifications to discern relationships. Impressively, the author list and corresponding dataset curated warrants acknowledgement for such a collaborative effort affords a unique opportunity to ask very interesting questions. Ultimately, the major conclusions posited from this work include a highly varied responses across population highlighting the importance of trophic guild and local context in human modification index as well as an increase in nocturnality. Unfortunately, more study justifications needed as the novelty of work is not convincingly presented. There have been so many COVID investigation studies including many cited in the text. Even with shortcoming discussed in the introduction section, results from this present study do not deviate from previous reports. As such, the added value with perhaps more data at a broader scale is not revealing much new insights. Additionally, there are many major shortcomings that hinder providing an overwhelming positive review with methodological concerns compromising the rigor and interpretation of findings. Minor comments are also provided to help improve the manuscript in future revisions.

7MAJOR ISSUES

>Human Modification Index (HMI) is misinterpreted and wrongly casted throughout the paper. This composite metric includes croplands as a variable. Therefore, its incorrect to state that this index is reflecting an urban-rural gradient, as certain locations with high HMI could reflect high agricultural production lands.

>Timing of human activity not incorporated into paper's investigation, despite making such a distinction with mammal responses in that timing (i.e., nocturnality) vs detection rate represents different variables of importance. As such, its odd that timing of human activity is stated as a possible policy intervention to promote human-wildlife coexistence when effects should have been tested first in the study (Line 476).

>The author stress limitations of previous works using "indirect measures of human activity" (Line 288) and then repeat same offense in their methods (e.g., "insufficient human detections from cameras", we used other data or local knowledge of changes in human activity (e.g., lockdown dates, visitor use data"): Supp Lines 119-20. There were other concerning designations as well that compromised study rigor (e.g., data contributors specifying population hunting status).

>It is not evident how "standardized" the studies included were, even after reviewing Wildlife Insights Metadata and contribution criteria. An additional table is needed in the supplement that provides - camera type (e.g., flash camera), sampling effort, number of stations, dates of operations, etc. Also unclear whether surveys were baited or unbaited, camera placement (e.g., on trails, roads).

>The incorporation of species traits seems over sold and in actuality less informative than presented. Namely because across the wide range for some species, species traits simply represent averages while populations exhibit extreme variation. As hunting was considered at the population level, many other traits should have been also (although, I recognize the challenge in implementing such given scale of project).

>Differences in study design comparison problematic that the pandemic reference varied in being "control" for within-year comparisons but was the designated as "treatment" for between year comparisons. Furthermore, this comparison type was not well presented in main text results for activity but was for timing of activity.

>It is unclear if sample sizes were sufficient to accurately evaluate diurnal species with only 16 species compared to nocturnal species including 60. Therefore, without knowing the distribution of these species across high and low activity sites/periods, difficult to interpret increase in nocturnality results.

>For model selection, AICc weights and an estimate of model fit are needed to ensure proper interpretation.

>Lastly, the exclusion of domestic species not only represents a missed opportunity but potentially confounds results depending on how tightly correlated they are with human presence. Given competitive interactions between domestic ungulates and carnivores with wild ungulates and

carnivores, by not accounting for their presence the effects of humans on mammals could be over or underestimated.

MINOR ISSUES

Unclear in intro casting of hypothesis, if generalist and specialist is referring to habitat, diet, or both (Lines 276-278). Also in this section, no justification for brain size being an important trait to include. And the type of human activity not included in study but stated here as being important.

Table 1: correlation among traits not stated in supplement, predictions not fully consistent with questions being investigated e.g., trophic and body size effects on response human activity given human modification levels. Also, designation of small carnivore is not consistent with the literature as Marneweck et al. 2021 Bio Con recent small carnivore review used < 16kg

Writing style/paragraph structure – for both paragraphs beginning at Line 391 and L 428, topic sentence does not match paragraph content. Therefore, it reads like a catch-all paragraph without clear focus.

The species variation was amongst the most interesting finding and warrants developing this story, potentially as case studies. Adding variation in activity and nocturnality to Table S1 would also help

Major sample size differences for changes in activity with 1065 populations then more than halved for timing of animal activity (n=499 populations) warrants explanation

Unclear if the 30 minute quiet period applied to human detections as well. Not convinced the same rationale would apply, and a shorter period may be more appropriate depending on camera site locations.

Fig 1: unclear vertical dashed; in B misaligned with end of the year

Fig 2, 3 for models – unclear why large herbivore was used as the reference trophic group

Line 315-317 - while true in mimicking the global domination of humans, from a management and mitigation perspective more insightful to consider if decreasing human activity is consequential.

Reviewer #3 (Remarks to the Author):

I agree with the authors that the "anthropause" during the Covid19 global pandemic and the ongoing passive monitoring projects using camera traps provided an excellent opportunity to conduct a control-treatment study of how mammals respond to human activity. Evidence is mounting about inter- and intra-specific variations in responses to changes in habitat. The manuscript is straight to the point that some species are more sensitive than others facing global change. I have a few methodological reservations and suggestions to clarify the analytical choices and the discussion of the findings. Specific comments are below.

(1) Line 314 - The definition of change in human activity could be quantified within this study. Was there a threshold defined for the direction of the effects as increasing, decreasing vs no change? I find this description a bit subjective for selecting datasets, and it might be worth reporting any quantification that has been performed to decide which projects/datasets to include.

(2) To my understanding, human activity (photo capture rate) is included as random intercepts to model animal activity. But was this not the gist of the analyses to quantify how animal activity varies with human activity? Why not include the effect of human activity as a fixed effect? How does human activity correlate with other anthropogenic covariates in Fig S2? Line 342-344 (supplementary text) includes interpretation of the results based on human activity and human modification index. I wonder if these were based on an interaction coefficient.

(3) The total amount of human activity is one aspect of human disturbance. A lot of times, spatial units within which animals move are limited, and temporal partitioning allows animals to minimize risks. In locations with high human activity (many photos of humans), how did animal detections follow/avoid human detections? Including information on time-since-human detections would provide a finer lens to look at how animals responded. This information could be the response variable within a survival analysis framework (using a Cox proportional hazard model), a time-to-even occupancy analysis, or a covariate on detection rates.

(4) The magnitude of human presence might not be captured through the number of photos taken by camera traps because people may see or be informed about the presence of the camera and actively avoid it. This is especially dependent on the camera trap setup and major human activities nearby. I am not sure how spatial variability and autocorrelation in human detection rates are accounted for.

(5) The comparison between animal detection rates during treatment vs. control (Line 165) is not reported in Table S2. From the caption, the change in detection rates was the response (not the detection rates). I wonder why the effect of human activity is quantified without having a covariate for treatment vs. control or a covariate for human capture rates. What is the benefit of a two-stage analysis compared to modeling all covariates effects in one model?

(6) How correlated were the species' traits? I wonder if one species' trait could have masked the effect of others.

(7) For figures with covariate values throughout the manuscript and supplementary materials, it would be good to back-scale and show the actual covariate values.

(8) Table S1. How about the range of human capture rates across the locations with detections of each species? Was there enough variation in the range of anthropogenic covariates to explain the variation in animal capture rates for all the species? I see that the range of covariate values is summarised in Table 1. I appreciate a similar summary per species.

(9) It'd be good to discuss the fact that the dataset, although comprehensive, is biased towards North America and Western Europe. There were considerable differences in regional policies in response to the global pandemic (e.g., restriction of human activity), and patterns observed in this study are more

10

relevant to North American and European mammalian communities and not necessarily a global response as phrased in the title/manuscript.

(10) Line 157-160. The detection rate is not the most reliable measure of habitat use (cautioned previously in the literature, e.g., Sollmann et al. 2013). I wonder why a hierarchical modeling framework is not used to quantify habitat use.

(11) At fine spatial and temporal scales, species assemblage could affect animal activity/habitat use. I understand that the study aimed to quantify the human impact, but it is worth mentioning that the wildlife communities present at treatment vs. control were similar, and species interactions were expected to be similar.

(12) Line 342-343. I am missing a quantification of intra-specific variations. For species with several datasets/projects, how did these responses vary? Was there a standard deviation that indicated spatial variations or spatial covariates that explained variations in intraspecific responses to human activity?

(13) Human modification index is a composite variable consisting of several stressors that may go beyond the binary of urban vs. rural areas. Low HMI was interpreted as rural areas (compared to high HMI as urban areas), which might not always be correct.

(14) I appreciate a discussion of all of the negative results. For example, the hypothesis about diurnal species being more sensitive to increased human activity makes sense, but how the lack of evidence is interpreted? Not having enough data in the species trait category might be part of the issue (only 16 diurnal species vs. 147 in the other two activity time categories).

Mahdieh Tourani

*****END*****

Author Rebuttal to Initial comments

Reviewers' comments:

Reviewer #1 (Remarks to the Author):

11Review for:

Mammal responses to global changes in human activity vary by trophic group and landscape

NOTE: I have written this review in markdown format. I also attached a PDF if that is easier for you to read.

In this paper the authors amassed a huge collection of camera trap data and used a quasi-experimental approach to assess how mammals modified their relative activity patterns and diel behavior in response to a shift in human activity caused by COVID-19 lockdowns. Overall, this paper was an absolute pleasure to read. The writing was fantastic, the figures are solid, and the story that is told from these data is very interesting. Perhaps some of the most novel results from this analysis is the shifts in activity patterns as a function of human modification and trophic guilds. Given the scale of the data, this finding represents very solid evidence that mammal responses substantially vary depending on their trophic guild in predictable in understandable ways. As such, I think this paper would be of interest to a broad readership.

Thank you Dr. Fidino for your kind words and constructive review!

Based on my reading of the paper, I have two larger concerns.

1. Based on reading the paper (and not the methods) a reader does not especially know if the landscape variation the paper assesses is within or among different camera trap studies. Being more specific about this would really help given that it helps the reader understand the scale at which inference is being made. In fact, given that the scale does appear to be at the project level (now that I have read them), there should likely be some reference to how the results here represent larger-scale averages right? I'm thinking that even within a camera trap study there is going to be variation (e.g., along an urban gradient within a study) and this variation is not being assessed, which is totally fine for this scale of

12analysis. However, I can easily see readers getting confused about how to interpret the overall effects here (i.e., interpret them at the camera trap level instead of the project level) which is not ideal. Providing some nuance on this, as well some appropriate caveats about what this analysis therefore represents, would help.

Thank you for catching this. We have clarified that our analysis assesses the effects of landscape-level variation among projects (sites), rather than within them (cameras). Specifically, in the main text, we added that we examined site-level changes (line 319) and variation in animal responses across sites (line 323) (note that we define the term site as referring to the project scale on line 312). On line 356 we emphasize that HMI was measured at the site (project) level.

In the Supplementary Information, we also added "site-level" for clarity (line 95), and our assumptions about site- vs. camera-level variation are made clear on lines 138-143.

2. From my reading of the methods (and the fact that code was not provided) I could not figure out what the uncertainty metric is that got created for the risk ratios for the nocturnality analysis. It seems like they exist (I discuss this a bit in the methods section at the bottom of the review) but that was not made clear in the supplemental methods section. Essentially what I want to make certain of is that there is some weighting by sample size that is occurring given that there is likely going to be some large scale variation in sample size across species and projects. Proportions generated from thousands of records should have more weight / more certainty than those generated from just a handful.

You are correct about weighting. The sampling variances derived from the risk ratios and the sample sizes are included in the effect size calculation. We used the default method in the 'metafor' package, which relates to equation 1 in Hedges et al. "The meta-analysis of response ratios in experimental ecology." Ecology 80.4 (1999): 1150-1156. We have clarified this point on lines 188-190 of the Supplementary Information.

We have provided our R code and analysis dataframes with this revised submission; the FigShare links are added to the Data Availability statement on lines 668-671 of the main text.

These issues, however, seem like they can be addressed with just a small amount of revision of the text.

Great work on this, if you have any questions about the comments here feel free to reach out to me at mfidino@lpzoo.org. I have comments for each section below as well, I hope you find them useful.

- Mason Fidino

Abstract

Top-level thoughts

Only thing I think I would add into this abstract is a tiny peppering of quantitative results. For example, the abstract states mammals were less active in rural areas under higher human activity. On average, what was the magnitude of this relationship? Adding something like `mammals were, on average, X times less active in rural areas...` would be welcome (though I suppose how representative an average like this across so many species and projects is something to consider as well). Overall , great synopsis!

You are correct that stating the average does not adequately capture the nuance of our results, and we feel that word limits within the abstract are too tight to accommodate such quantitative descriptions. However, to address this (and your comment below) we now present the predictions from the models in terms of % change of our response terms, which we feel is more interpretable (see updated Figure 2D; Figures 3D+3E; and Figures S3+S4; SI results). We also added these quantitative summaries of % changes throughout the main text results (e.g., lines 356-7), and we added a brief description of the calculation of these summaries to the SI Methods under the Meta-analysis models (SI lines 313-316).

14Introduction

Top-level thoughts

1. The second paragraph uses `animals` but in reality I think the authors are more specifically referencing `mammals`. For example, birds of larger size are actually more tolerant of human presence, not small-bodied (Neate-Clegg et al. 2023). This is in opposition to the point the authors make on lines 276-278. I'm splitting hairs here, but given that the title of the manuscript has already brought the focus down to mammals I think being more specific here would be warranted.

You are correct in noting that our paper focuses on mammals (as highlighted in title, abstract, etc). But with that said, we see value in keeping our introduction broad. Furthermore, we had the opposite interpretation of the Neate-Clegg et al paper, which states in the abstract that “urban-associated species tend to be smaller”, thus our statement should be broadly applicable to more animals than only mammals.

Montague H.C. Neate-Clegg, Benjamin A. Tonelli, Casey Youngflesh, Joanna X. Wu, Graham A. Montgomery, Çağan H. Şekercioğlu, Morgan W. Tingley. (2023).

Traits shaping urban tolerance in birds differ around the world,

Current Biology

<https://doi.org/10.1016/j.cub.2023.03.024>.

Line by line comments

15Line 292-305: Great points here.

Changes in amount of animal activity

Top-level thoughts

1. At this stage I am not sure what scale these regression coefficients are, and as a result I am uncertain of the magnitude of effect when they are reported. As just one example, I am unsure how big the effect size really is for the results on lines 354 - 357. How much did animal activity increase with higher levels of human activity? Figure 2C does not help with this either. In fact, the whole paragraph on lines 354 - 375 only references direction of effect without providing any reference to how large this differences are. To me, this appears to be a really big missed opportunity based on those slopes in Fig. 2D and raw treatment effects on Fig 2.B. Certainly there are some space constraints, but it would be great to add some more quantitative statements about these results in this section (because it seems like these responses are quite large for some species).

As noted above, we now present the predictions from the models in terms of % change of our response terms (see updated Figure 2D; Figures 3D+3E; and Figures S3+S4; SI results section). We also added these quantitative summaries of % changes throughout the main text results (e.g., lines 356-7).

Line by line comments

Line 346: Any uncertainty in that estimate (i.e., -0.04)?

We added the 95% confidence intervals around the mean (95% CI = -0.11 - 0.03)

Line 346 - 349: This sentence is a little bit confusing because the reference to the response variable (activity) happens quite far into the sentence. This makes the first part of the sentence, before the comma, a little confusing. What does it mean that trophic ground is a predictor of responses to increasing human use? A bit more clarity here would help the reader, especially because the methods are supplemental. However, I am not a personal fan of calling out something as confusing without at least attempting a suggested revision. Maybe something like this?

...

How animal activity rates changed in response to increasing human use varied by trophic group (combining body mass and trophic level), with large herbivores showing the largest increases in activity and carnivores the strongest decreases (Fig. 2C, etc. etc.).

...

Thank you for the suggestion. We clarified the sentence by adding “strongest predictor of changes in animal activity in response to increasing human use”

Line 350: maybe use `mortality risk` instead of $\text{`risk of mortality`}$ so you don't have two of's chained together.

We made the suggested change.

Line 354-357: Is this at the project level or within a project? In other words, is this variation along a gradient of human modification within a camera trap study or is it variation in the average human modification index across different camera trap studies?

We added “measured at the site level” to clarify (having defined project sampling areas as sites earlier)

Changes in timing of animal activity

Top-level thoughts

1. Same comment applies here as above. Some more quantification of effect size in the text would help.

As noted above, we now present the predictions from the models in terms of % change of our response terms (see updated Figure 2D; Figures 3D+3E; and Figures S3+S4; SI results section). We also added these quantitative summaries of % changes throughout the main text results (e.g., line 413).

Implications for human-wildlife coexistence

Top-level thoughts

1. The take-away from the paragraph on lines 453 - 467 hinges on whether this among-species average has meaning. It means very little when you focus your attention to large-carnivores, for example. I think

you are providing that nuance in the last line of the paragraph, but not sure if `...certain species and contexts...` is an explicit call out to this?

We believe that the among-species average does have meaning. As we allude to in the paragraph, a popular narrative during this period of the pandemic was that all animals increased their activity when human activity decreased (and by implication, vice versa), which would have resulted in a mean change across all populations that was very far from zero, contrary to what we observed.

And yes, our closing sentence is meant to set up the transition to the following paragraphs, in which the significant differences among contexts and species are discussed.

Methods

NOTE: This is by line number within the supplemental material

Top-level thoughts

1. For the timing of animal activity. Is there any uncertainty associated to the log risk ratio? It would seem to me that there should be less weight for deployments / species with less data but from my reading of 174 - 187 it seems like every data point would be equally weighted in subsequent analyses. To me, it would seem like you would want to

a) Fit some type of Bayesian analysis (e.g., logistic regression) to get the estimated proportion of nighttime animal activity at high and low human activity levels. This would result in a posterior, so you would calculate your risk ratio with uncertainty.

19b) Fit a frequentist logistic regression but do some bootstrapping. This would also result in a distribution of proportions.

In my opinion, the Bayesian analysis would be easier than a bootstrap, especially if you need to add a random effect. However, it does seem like there is some uncertainty that was created given that Fig 3B has 95% CI? If there is uncertainty then that should be made more clear here as I am uncertain what the sampling variance is that gets provided into the meta-analytic model.

The sampling variances derived from the risk ratios are included in the meta-analysis models. We used the default method in the 'metafor' package (which relates to equation 1 in Hedges et al. 1999 "The meta-analysis of response ratios in experimental ecology." Ecology 80: 1150-1156). We have clarified this point on lines 190-192 of the Supplementary Information.

2. The justification for not accounting for variation in sampling among projects is confusing (lines 193 - 196) given that the covariates that got calculated represent differences in sampling among projects (i.e., where cameras were placed). Do the authors instead mean variation in study design for projects (e.g., how high cameras were placed, if lure was used, etc.)?

We added sampling "protocols" and "e.g., camera placement and settings" to clarify what we meant (lines 201-204).

3. When calculating independent detections it appears that this was done at the species and camera level. However, it does not appear that there was any checking to determine if cameras were independent of one another? Some studies could have had paired cameras, and so the same individual could be getting captured on two "independent cameras." I'm assuming the authors accounted for this, but adding a sentence in the methods here with respect to this would help. Likewise, it could help to report the minimum distance between cameras (average across studies, min and max).

We added the following sentence (SI lines 90-92) to clarify that cameras were not paired:

“Camera locations were considered independent within projects, as no paired cameras were included (see Table S10 for more details on camera spacing).”

We provided the spacing (mean = 2.00 km; min = 0.03 km; max = 26.20 km) and other project-level summaries in a new supplementary Table S10.

4. A sentence on what the human modification index represents would help with the interpretation of your results. I had to google it to find out what it represented.

We added a statement on lines 257-9 to clarify that HMI “represents a cumulative measure of the proportion of a landscape modified by 13 anthropogenic stressors”

Line by line comments

Line 156-157: Was sampling effort a log offset term or just included in the model? I'm assuming an offset (as it should be) but that information is left out. Wait, I see it now on lines 164 (it's an offset), just hint at it here and you are golden.

We added “offset” here as well (line 163)

NOTE: I added my name above, but the reviewer instructions also asked me to put it here.

Mason Fidino

Reviewer #2 (Remarks to the Author):

The authors here investigate the effects of human activity associated with 2020 COVID pandemic on mammal activity and nocturnality. They integrate several explanatory variables based primarily on species traits and interactions with habitat and human modifications to discern relationships. Impressively, the author list and corresponding dataset curated warrants acknowledgement for such a collaborative effort affords a unique opportunity to ask very interesting questions. Ultimately, the major conclusions posited from this work include a highly varied responses across population highlighting the importance of trophic guild and local context in human modification index as well as an increase in nocturnality. Unfortunately, more study justifications needed as the novelty of work is not convincingly presented. There have been so many COVID investigation studies including many cited in the text. Even with shortcoming discussed in the introduction section, results from this present study do not deviate from previous reports. As such, the added value with perhaps more data at a broader scale is not revealing much new insights. Additionally, there are many major shortcomings that hinder providing an overwhelming positive review with methodological concerns compromising the rigor and interpretation of findings. Minor comments are also provided to help improve the manuscript in future revisions.

Thank you for recognizing the unique contribution of this large collaborative effort, and for your thorough review. We respectfully disagree that our study is the same as previous investigations of wildlife responses to COVID-related changes in human activity. Firstly, we frame our study as not simply a “COVID” paper, but rather as a more general evaluation of animal responses to changes in human activity—particularly since our study highlighted such large variation in human activity during the “lockdown” periods (lines 325-9). Importantly, we emphasize in several places (e.g., lines 296, 309) that there have not been previous studies that match the spatial scale and sample size of our assessment using a standardized, rigorous methodological framework. Rather, previous studies have either been isolated case studies (single sites or species) or large-scale syntheses using disparate methods with uncertain biases on inferences. Finally, we are not aware of any previous studies that have our two main results: the greater sensitivity of carnivore species and the importance of landscape context (i.e., human modification). These points are highlighted in several areas (e.g. abstract, lines 472-5, 490-3).

MAJOR ISSUES

22>Human Modification Index (HMI) is misinterpreted and wrongly casted throughout the paper. This composite metric includes croplands as a variable. Therefore, its incorrect to state that this index is reflecting an urban-rural gradient, as certain locations with high HMI could reflect high agricultural production lands.

It is correct that the HMI includes croplands as well as 12 other anthropogenic variables in a composite measure of human modification of the landscape, which we interpret as a measure of anthropogenic disturbance as intended. As noted on line 261 of the SI, HMI was highly correlated with human population density across our sampling sites, thus we suspect that it does primarily reflect a more urban-to-rural gradient (terminology we originally chose to appeal to broad readership). However, to avoid confusion and address your concern, we replaced “urban” with “developed” or “more developed” and rural with “undeveloped” or “less developed” throughout the manuscript (and we removed our literary reference to Aesop’s fable of the Town Mouse and Country Mouse).

>Timing of human activity not incorporated into paper’s investigation, despite making such a distinction with mammal responses in that timing (i.e., nocturnality) vs detection rate represents different variables of importance. As such, its odd that timing of human activity is stated as a possible policy intervention to promote human-wildlife coexistence when effects should have been tested first in the study (Line 476).

We used human activity to define our treatment (increase in human activity) and as a predictor variable (magnitude of change). It is generally established that human activity is predominantly diurnal in terms of interactions with wildlife, and there is evidence of wildlife shifting toward greater nocturnality as a response to avoiding human activity (Gaynor et al. 2018, as cited in our manuscript). It would be interesting to conduct a follow-up analysis of potential shifts in timing of human activity, but that is beyond the scope of our paper. We mention management of the timing of human activity in Discussion as an idea for managers to consider in the future, as we anticipate that with growing human activity, it could be possible that more of that activity occurs at night.

>The author stress limitations of previous works using “indirect measures of human activity” (Line 288) and then repeat same offense in their methods (e.g., “insufficient human detections from cameras”, we used other data or local knowledge of changes in human activity (e.g., lockdown dates, visitor use

data”): Supp Lines 119-20. There were other concerning designations as well that compromised study rigor (e.g., data contributors specifying population hunting status).

We appreciate this concern. We only used other data or local knowledge in 15 of 102 cases where our empirical (camera) data were insufficient, so we used the empirical data in 85% of cases. And we were critiquing coarse measures of mobility (e.g. county-level indicators) or data sources known to be biased (e.g. ebird), as opposed to expert local knowledge.

Unfortunately there are no other data readily available on hunting status, and we value and trust the local knowledge of our contributing experts, who know their study systems well. We would consider any constructive suggestions for improvement.

>It is not evident how “standardized” the studies included were, even after reviewing Wildlife Insights Metadata and contribution criteria. An additional table is needed in the supplement that provides - camera type (e.g., flash camera), sampling effort, number of stations, dates of operations, etc. Also unclear whether surveys were baited or unbaited, camera placement (e.g., on trails, roads).

Since our comparisons of the effects of changes in human activity are within projects (i.e., paired treatment vs. control in the same sampling area), we feel that potential between-project variations in specific camera protocols should have minimal influence on our results. Nevertheless, to improve transparency of the methods, we added a new supplementary table (Table S10) detailing the project name, information, number of unique stations included in the analysis, average spacing (mean distance to the nearest station), bait use (only 8 of 102 projects) and the % of cameras baited, treatment designations, and dates chosen and duration of each period by treatment type.

>The incorporation of species traits seems over sold and in actuality less informative than presented. Namely because across the wide range for some species, species traits simply represent averages while populations exhibit extreme variation. As hunting was considered at the population level, many other traits should have been also (although, I recognize the challenge in implementing such given scale of project).

While we appreciate the general importance of considering within-species variation in traits, we feel this is a criticism that could be leveled at 1000s of papers that use species-level traits. In the absence of a population-level trait database, we are left with species-level values, which still revealed interesting results in our case.

>Differences in study design comparison problematic that the pandemic reference varied in being “control” for within-year comparisons but was the designated as “treatment” for between year comparisons. Furthermore, this comparison type was not well presented in main text results for activity but was for timing of activity.

Our experimental “treatment” for both models was higher human activity. Whether this occurred during the pandemic period or not varied by project and was not related to whether it was a within-year or between-year comparison. We feel this is clearly explained on lines 317-319, as follows:

“[We] standardized our comparisons to be between periods of relatively lower to higher human activity (either across years or within 2020; Fig, 1; SI Methods) to mimic the general trend of increasing human presence in the Anthropocene.”

>It is unclear if sample sizes were sufficient to accurately evaluate diurnal species with only 16 species compared to nocturnal species including 60. Therefore, without knowing the distribution of these species across high and low activity sites/periods, difficult to interpret increase in nocturnality results.

All of the populations included were subject to a low-vs.-high contrast in human activity. Furthermore, we make no firm statements about diurnal species as the confidence intervals around that effect are broad (Figure 3C), which could reflect highly variable responses within this group, or, as you state, low sample size. We already have a column in SI Table 1 showing the number of populations sampled for each species (i.e., project-species); we have now added an additional column showing the range of human modification values that these populations cover.

>For model selection, AICc weights and an estimate of model fit are needed to ensure proper interpretation.

We added AICc weights to model selection tables (Tables S3-S5, S7-S9) and added a pseudo-R² metric in the SI text which reports the proportional change in the variance components when adding the fixed effects (R² for full models: amount of activity = 25.4%; timing of activity = 30.2%).

>Lastly, the exclusion of domestic species not only represents a missed opportunity but potentially confounds results depending on how tightly correlated they are with human presence. Given competitive interactions between domestic ungulates and carnivores with wild ungulates and carnivores, by not accounting for their presence the effects of humans on mammals could be over or underestimated.

The combined detections of domestic animals (dogs, cats, cows, horses, sheep, goat, donkey, domestic rabbit) represented a small proportion (6%) of the overall detections in our dataset, and half of these were detections of dogs. Since dogs (and many domestics) are typically associated with humans, we did not consider them separately (we added text to SI lines 85-86 to clarify this). While we agree that further analysis of relationships between wild and domestic animals is interesting, we feel it would require a separate analysis that is beyond the scope of our paper (we do not have an a priori hypothesis for changes in domestic animals during Covid that would affect wild animals independent of changes in human activity).

MINOR ISSUES

Unclear in intro casting of hypothesis, if generalist and specialist is referring to habitat, diet, or both (Lines 276-278). Also in this section, no justification for brain size being an important trait to include. And the type of human activity not included in study but stated here as being important.

We mean generalist to refer to both habitat and diet, which we feel is reflected in our reference later in the sentence to “shifting resource use within their broader niches”.

26In Table 1 and the SI (under Species traits, lines 207-210), we explain our hypothesis that species with smaller relative brain sizes will be more sensitive to changes in human activity (with supporting citation). We did not have space in this section of the main text to provide detailed justifications for all traits/variables tested.

While we do not look in depth at the effects of different types of human activity, we did in fact assess the effects of hunting in our analysis, which is related to the example given in this part of the main text (“hunting vs. hiking”).

Table 1: correlation among traits not stated in supplement, predictions not fully consistent with questions being investigated e.g., trophic and body size effects on response to human activity given human modification levels. Also, designation of small carnivore is not consistent with the literature as Marneweck et al. 2021 Bio Con recent small carnivore review used < 16kg

Correlations between continuous traits are shown in Figure S2B.

We picked 20 kg as a threshold as a compromise between all three groups (herbivore, carnivore and omnivore).

Writing style/paragraph structure – for both paragraphs beginning at Line 391 and L 428, topic sentence does not match paragraph content. Therefore, it reads like a catch-all paragraph without clear focus.

Both of these paragraphs represent additional results (for amount and timing of activity, respectively) in the sense that they combine statements pertaining to other variables tested in our models. While it would be possible to add a general topic sentence such as “Other variables tested showed mixed results”, this seems to us self-evident and inefficient for a manuscript with such tight word limits. Since each of

these paragraphs follow two others which detail the primary results for each response variable, we feel it is reasonable to leave the paragraphs as they are.

The species variation was amongst the most interesting finding and warrants developing this story, potentially as case studies. Adding variation in activity and nocturnality to Table S1 would also help

We added the range of raw response effects sizes for each species into Table S1 as requested. Unfortunately the word limits for the manuscript prevent us from including several case studies for individual species (although we do highlight several species-specific results, such as on lines 364-376 of main text).

Major sample size differences for changes in activity with 1065 populations then more than halved for timing of animal activity (n=499 populations) warrants explanation

We applied a more stringent threshold of 10 detections within both treatment and control periods to calculate the timing of activity, in order to ensure that we had sufficient detections to reasonably estimate changes in diurnality vs. nocturnality. We have clarified this on lines 188-9 of the supplementary information.

Unclear if the 30 minute quiet period applied to human detections as well. Not convinced the same rationale would apply, and a shorter period may be more appreciate depending on camera site locations.

We now clarify that we also used the number of independent human detections, in the same way as done in the animal analysis (lines 96-7 of SI). This does not influence the results as we used human detections as an index of human activity (rather than absolute number of humans), and this index is standardized between the low and high periods of comparison.

Fig 1: unclear vertical dashed; in B misaligned with end of the year

We removed these lines as they did not aid interpretation.

Fig 2, 3 for models – unclear why large herbivore was used as the reference trophic group

This is an arbitrary choice (one group had to be chosen) but should not affect interpretation of results.

Line 315-317 - while true in mimicking the global domination of humans, from a management and mitigation prospective more insightful to consider if decreasing human activity is consequential.

This point could be fairly argued both ways, and we had to choose one.

Reviewer #3 (Remarks to the Author):

I agree with the authors that the "anthropause" during the Covid19 global pandemic and the ongoing passive monitoring projects using camera traps provided an excellent opportunity to conduct a control-treatment study of how mammals respond to human activity. Evidence is mounting about inter- and intra-specific variations in responses to changes in habitat. The manuscript is straight to the point that some species are more sensitive than others facing global change. I have a few methodological reservations and suggestions to clarify the analytical choices and the discussion of the findings. Specific comments are below.

Thank you Dr. Tourani for your thorough review and feedback.

(1) Line 314 - The definition of change in human activity could be quantified within this study. Was there a threshold defined for the direction of the effects as increasing, decreasing vs no change? I find this description a bit subjective for selecting datasets, and it might be worth reporting any quantification that has been performed to decide which projects/datasets to include.

We distinguished increasing/decreasing/same based on detection rates of humans by camera traps, and we confirmed with data contributors that the classification was consistent with local knowledge of the system. In cases where there were insufficient human detections, we relied on expert opinion (only 15 of 102 projects, noted in SI). Although we did not establish a quantitative threshold for defining “change”, we included the estimated magnitude of change in human activity (from camera detections) as a covariate in models for all projects where this was available (see Tables S5 and S9), which we feel accounts for the variation among projects within the same category of direction of change.

(2) To my understanding, human activity (photo capture rate) is included as random intercepts to model animal activity. But was this not the gist of the analyses to quantify how animal activity varies with human activity? Why not include the effect of human activity as a fixed effect? How does human activity correlate with other anthropogenic covariates in Fig S2? Line 342-344 (supplementary text) includes interpretation of the results based on human activity and human modification index. I wonder if these were based on an interaction coefficient.

Human activity was not included as a random intercept. The random intercepts in our models were project ID (as we have multiple species from the same project) and family/species (to account for multiple observations from the same taxonomic groups) - as stated on lines 287-9 of the SI.

Human activity was included as a fixed effect in models run on projects for which this empirical measure was available (see tables S5 and S9). We added the correlation between camera derived change in human activity with our other indices of human disturbance as a second panel to Figure S1.

(3) The total amount of human activity is one aspect of human disturbance. A lot of times, spatial units within which animals move are limited, and temporal partitioning allows animals to minimize risks. In locations with high human activity (many photos of humans), how did animal detections follow/avoid

30human detections? Including information on time-since-human detections would provide a finer lens to look at how animals responded. This information could be the response variable within a survival analysis framework (using a Cox proportional hazard model), a time-to-event occupancy analysis, or a covariate on detection rates.

A time-to-event analysis is one of several interesting finer-scale extensions that we hope to explore with this dataset in future papers, but that are beyond the scope of this paper.

(4) The magnitude of human presence might not be captured through the number of photos taken by camera traps because people may see or be informed about the presence of the camera and actively avoid it. This is especially dependent on the camera trap setup and major human activities nearby. I am not sure how spatial variability and autocorrelation in human detection rates are accounted for.

We requested all data contributors to confirm that the observed change in human detections on camera traps was a reliable indicator of change in human activity within their study areas (i.e. not likely to be due to some form of detection bias as you suggest). Furthermore, our paired treatment-control comparisons within projects (same cameras in the same locations in each period) controls for this.

(5) The comparison between animal detection rates during treatment vs. control (Line 165) is not reported in Table S2. From the caption, the change in detection rates was the response (not the detection rates). I wonder why the effect of human activity is quantified without having a covariate for treatment vs. control or a covariate for human capture rates. What is the benefit of a two-stage analysis compared to modeling all covariates effects in one model?

Conducting the two-stage analysis (i.e., estimate the treatment effect then test factors hypothesized to explain variation in effects) was necessary to adopt the meta-analysis framework allowing us to synthesize and test for general patterns across all projects. If we were only looking at one project, we would adopt the approach you suggest.

(6) How correlated were the species' traits? I wonder if one species' trait could have masked the effect of others.

We have already included a plot of the relationships between continuous traits in Figure S2B.

(7) For figures with covariate values throughout the manuscript and supplementary materials, it would be good to back-scale and show the actual covariate values.

Whilst we would typically agree, the continuous covariates that this would affect (i.e., human modification, diet/habitat breadth and lockdown stringency) are already abstract indices. As such, back-transformation to the "observation scale" doesn't really help interpretation.

(8) Table S1. How about the range of human capture rates across the locations with detections of each species? Was there enough variation in the range of anthropogenic covariates to explain the variation in animal capture rates for all the species? I see that the range of covariate values is summarised in Table 1. I appreciate a similar summary per species.

We do not have effect sizes for magnitude of change in human activity for all populations (since we could not estimate it for all projects). In its place, we added the range in human modification index values for each species to Table S1.

(9) It'd be good to discuss the fact that the dataset, although comprehensive, is biased towards North America and Western Europe. There were considerable differences in regional policies in response to the global pandemic (e.g., restriction of human activity), and patterns observed in this study are more relevant to North American and European mammalian communities and not necessarily a global response as phrased in the title/manuscript.

We added caveats to the main text ("predominantly in Europe and North America, line 312) and supplement ("These projects spanned 21 countries, mostly in North America and Europe but with some

32representation from South America, Africa, and Southeast Asia (Fig. 1, Table S10)”, lines 135-7) to acknowledge the uneven geographic coverage of our samples.

We had also previously called attention to the need to have “expanded assessment of contexts underrepresented in our sample, such as tropical regions subjected to different pressures during the pandemic” and to fill geographic gaps in biodiversity monitoring on lines 514-8 of the main text.

(10) Line 157-160. The detection rate is not the most reliable measure of habitat use (cautioned previously in the literature, e.g., Sollmann et al. 2013). I wonder why a hierarchical modeling framework is not used to quantify habitat use.

Sollmann et al. 2013 caution about the use of detection rates as indices of population abundance, a topic we have also investigated (e.g., Broadley et al. 2019, Ecology & Evolution 9:14031-14041). In this paper, we do not make inferences about abundance per se; rather we are using detection rates to quantify relative differences in animal activity (i.e., habitat use) within sites between the paired periods of high vs low human activity. Given the relatively short-term duration of “treatments” (i.e. before/during/after lockdowns), we do not interpret animal responses as changes in abundance, but as behavioral changes, which we measure with activity. Furthermore, our paired design (comparison across same cameras) controls for much variation that might affect detectability, making these relative changes more robust.

We assume that by hierarchical modeling framework you are referring to methods such as N-mixture or occupancy models that attempt to model detectability separately from count or occurrence. We argue that this is a topic of active research/debate and a matter of researchers assessing whether model assumptions are reasonable in the context of their study. Occupancy (and related) models have questionable assumptions in the context of many camera trap surveys of mammals (e.g. site closure, see for example Neilson et al. 2018, Ecosphere 9:e02092), and thus are not inherently superior to our modeling assumption that differences in detection rates reflect differences in animal activity.

(11) At fine spatial and temporal scales, species assemblage could affect animal activity/habitat use. I understand that the study aimed to quantify the human impact, but it is worth mentioning that the wildlife communities present at treatment vs. control were similar, and species interactions were expected to be similar.

We only made comparisons between periods of higher vs lower human activity at the same sites (in a similar season within the same year or separated by a year). Thus we have no reason to expect any differences in wildlife communities between periods, or any particular differences in species interactions that would influence our interpretations. The issue of how species interactions might change with changing human activity is an interesting one, but would require an additional analysis/paper beyond the scope of this one.

(12) Line 342-343. I am missing a quantification of intra-specific variations. For species with several datasets/projects, how did these responses vary? Was there a standard deviation that indicated spatial variations or spatial covariates that explained variations in intraspecific responses to human activity?

We calculate the I^2 statistic to estimate the amount of variation captured by the random effects: high values reflect random effects which explain large proportions of the variation (i.e., consistent differences between groups) and low values reflect the converse. For the models of amount of activity, only project ID consistently explained variation in responses, whereas in models of timing of activity (nocturnality), both project ID and species explained variation. We have reworded this to make it clearer (lines 363-9 and 405-8 of the SI).

(13) Human modification index is a composite variable consisting of several stressors that may go beyond the binary of urban vs. rural areas. Low HMI was interpreted as rural areas (compared to high HMI as urban areas), which might not always be correct.

We no longer use the “urban vs rural” language to describe high vs. low HMI (please see the detailed response above).

(14) I appreciate a discussion of all of the negative results. For example, the hypothesis about diurnal species being more sensitive to increased human activity makes sense, but how the lack of evidence is interpreted? Not having enough data in the species trait category might be part of the issue (only 16 diurnal species vs. 147 in the other two activity time categories).

We acknowledge that the lower sample size of diurnal species results in uncertain effects which are reflected in the manuscript (e.g., wider confidence intervals). As reflected in Table S1, we had a sample of 13 diurnal species; however, the sample size for the model was actually 32 populations of these 13 diurnal species, and though this is significantly lower than for the other activity classes (as you note), it still represents an unprecedentedly large synthesis in this context. We will have to leave it to future syntheses to increase that sample size.

Mahdiah Tourani

Decision Letter, first revision:

10th August 2023

Dear Dr Burton,

Your manuscript entitled "Mammal responses to global changes in human activity vary by trophic group and landscape" has now been seen by the same 3 reviewers, whose comments are attached. As you can see, although the reviewers are largely satisfied with the changes, they still raise some further concerns which will need to be addressed before we can offer publication in Nature Ecology & Evolution. We will therefore need to see your responses to the criticisms raised and to some editorial concerns, along with a revised manuscript, before we can reach a final decision regarding publication.

In particular, Referee #1 invites you to consider the use of phrasing related to covid-19, and Referee #3 would like to see some of the analytical choices justified in the methods, and a broader discussion included on the study's limitations. Please note that although Referee #2 chose not to pass on further comments, they expressed to me that although they still disagreed with some of the responses to their comments, they also felt that the study remained suitable for publication.

We therefore invite you to revise your manuscript taking into account all reviewer and editor comments. Please highlight all changes in the manuscript text file [OPTIONAL: in Microsoft Word format].

We are committed to providing a fair and constructive peer-review process. Do not hesitate to contact us if there are specific requests from the reviewers that you believe are technically impossible or

35unlikely to yield a meaningful outcome.

* If you have not done so already please begin to revise your manuscript so that it conforms to our Article format instructions at <http://www.nature.com/natecolevol/info/final-submission>. Refer also to any guidelines provided in this letter.

[REDACTED]

Nature Ecology & Evolution is committed to improving transparency in authorship. As part of our efforts in this direction, we are now requesting that all authors identified as 'corresponding author' on published papers create and link their Open Researcher and Contributor Identifier (ORCID) with their account on the Manuscript Tracking System (MTS), prior to acceptance. ORCID helps the scientific community achieve unambiguous attribution of all scholarly contributions. You can create and link your ORCID from the home page of the MTS by clicking on 'Modify my Springer Nature account'. For more information please visit please visit www.springernature.com/orcid.

[REDACTED]

Reviewers' comments:

Reviewer #1 (Remarks to the Author):

Great job on the revisions.

One last thing I was thinking about while reading this paper again is that it would show a bit of humanity here and acknowledge all of the people who have been impacted by the COVID pandemic. The text refers to the pandemic as an 'opportunity' and an 'unplanned experiment' and while those statements are true, it came at the cost of millions of human lives. As such, I think a little bit of care could be taken with respect to how the pandemic is referenced, either in the main text or perhaps as the only bit shared in the acknowledgements (and then link to full acknowledgements in supplemental material).

As a revision, this is of course minor, but I think it's an important piece to add. Again, great work on this, I look forward to citing it in the future.

Cheers,
Mason Fidino

Reviewer #2 (Remarks to the Author):

NA

Reviewer #3 (Remarks to the Author):

I have a follow-up comment about the range of covariate values that are provided in the revised version. For many species, there are limited to no variations in the human modification index (Table S1). How is the inclusion of these species justified in the analysis which explores the impact of that covariate? Are species listed in Table S1 all included in the analyzes?

*****END*****

Author Rebuttal, first revision:

Reviewers' comments:

Reviewer #1 (Remarks to the Author):

Great job on the revisions.

One last thing I was thinking about while reading this paper again is that it would show a bit of humanity here and acknowledge all of the people who have been impacted by the COVID pandemic. The text refers to the pandemic as an 'opportunity' and an 'unplanned experiment' and while those statements are true, it came at the cost of millions of human lives. As such, I think a little bit of care could be taken with respect to how the pandemic is referenced, either in the main text or perhaps as the only bit shared in the acknowledgements (and then link to full acknowledgements in supplemental material).

As a revision, this is of course minor, but I think it's an important piece to add. Again, great work on this, I look forward to citing it in the future.

Cheers,
Mason Fidino

We thank Mason for this important point. Following the suggestion, we added this statement to the main text Acknowledgements: "We recognize the tragic consequences of the COVID-19 pandemic and would like to acknowledge all people impacted."

Reviewer #2 (Remarks to the Author):

NA

Reviewer #3 (Remarks to the Author):

I have a follow-up comment about the range of covariate values that are provided in the revised version. For many species, there are limited to no variations in the human modification index (Table S1). How is the inclusion of these species justified in the analysis which explores the impact of that covariate? Are species listed in Table S1 all included in the analyzes?

Species listed in Table S1 were included in at least the analysis of changes in the amount of activity ("NA"

38in the second column indicates cases where a species was not included in the analysis of changes in timing of activity). It is true that some species are represented by only 1 population in our sample (i.e., detected in 1 project), and thus do not extend across variation in the landscape covariates such as the Human Modification Index (HMI). This is shown clearly in the first 2 columns of Table S1 (# of projects in which each species is represented for the two analysis response variables). However, the majority of species are represented in the sample by multiple populations (e.g. 105 of 163 species for amount of activity).

Importantly, our unit of analysis for assessing the effects of HMI was population (i.e., species-project combination), not species. This was explained in the manuscript on (e.g.) lines 343 and 404, and for this revision we added “populations of” as further clarification on line 319. We included species traits to test for general effects of traits shared across multiple species, and included the Species random effect to account for species with multiple populations, as well as a Family random effect for multiple species within the same family. Given this model structure (all explained in the text), we do not feel that having some species with only 1 population is a problem for our analysis. Furthermore, in a global-scale study like ours, it is not unexpected that some species would have only 1 or few populations included (given extensive sampling coverage rather than intensive within more restricted ranges). Finally, one of our interpretations (lines 480-482) is that sensitive species are likely to be “filtered” out of more highly modified landscapes, which implies that not all species would be expected to have populations across a wide range of HMI values.

Decision Letter, second revision:

14th December 2023

Dear Dr. Burton,

Thank you for your patience while we sought re-review comments on your revised manuscript "Mammal responses to global changes in human activity vary by trophic group and landscape" (NATECOLEVOL-23030556B), and my sincere apologies for the length of time it has taken to convey this decision to you. Unfortunately we were unable to secure the re-review of Referee #3, who had the last remaining comments, however one of the other Referees (Referee #1) has now stepped in to assess your response instead.

This reviewer finds that the paper has improved in revision, but they would still like to see the issue of low sample size bias addressed more explicitly in the final text. Specifically, they strongly recommend the inclusion of a simulation study in which the overall bias in species-level estimates with sample size if evaluated. Please find attached a Script and Figure which they suggested were passed on to you

39where they show that the bias in species-level estimates is strongly associated with sample size, while community-level inference is not affected.

At minimum, editorially we would like to see more explicit caution expressed in the text about making inferences on species with a small sample sizes. Once these additional changes are made we'll be happy in principle to publish it in Nature Ecology & Evolution, pending minor revisions to comply with our editorial and formatting guidelines.

[REDACTED]

Our ref: NATECOLEVOL-23030556B

3rd January 2024

Dear Dr. Burton,

Thank you for your patience as we've prepared the guidelines for final submission of your Nature Ecology & Evolution manuscript, "Mammal responses to global changes in human activity vary by trophic group and landscape" (NATECOLEVOL-23030556B). Please carefully follow the step-by-step instructions provided in the attached file, and add a response in each row of the table to indicate the changes that you have made. Please also check and comment on any additional marked-up edits we have proposed within the text. Ensuring that each point is addressed will help to ensure that your revised manuscript can be swiftly handed over to our production team.

**We would like to start working on your revised paper, with all of the requested files and forms, as soon as possible (preferably within two weeks). Please get in contact with us immediately if you

40anticipate it taking more than two weeks to submit these revised files.**

In recognition of the time and expertise our reviewers provide to Nature Ecology & Evolution's editorial process, we would like to formally acknowledge their contribution to the external peer review of your manuscript entitled "Mammal responses to global changes in human activity vary by trophic group and landscape". For those reviewers who give their assent, we will be publishing their names alongside the published article.

Nature Ecology & Evolution offers a Transparent Peer Review option for new original research manuscripts submitted after December 1st, 2019. As part of this initiative, we encourage our authors to support increased transparency into the peer review process by agreeing to have the reviewer comments, author rebuttal letters, and editorial decision letters published as a Supplementary item. When you submit your final files please clearly state in your cover letter whether or not you would like to participate in this initiative. Please note that failure to state your preference will result in delays in accepting your manuscript for publication.

Cover suggestions

We welcome submissions of artwork for consideration for our cover. For more information, please see our https://www.nature.com/documents/Nature_covers_author_guide.pdf guide for cover artwork.

Nature Ecology & Evolution has now transitioned to a unified Rights Collection system which will allow our Author Services team to quickly and easily collect the rights and permissions required to publish your work. Approximately 10 days after your paper is formally accepted, you will receive an email in providing you with a link to complete the grant of rights. If your paper is eligible for Open Access, our Author Services team will also be in touch regarding any additional information that may be required to arrange payment for your article.

Please note that Nature Ecology & Evolution is a Transformative Journal (TJ). Authors may

41publish their research with us through the traditional subscription access route or make their paper immediately open access through payment of an article-processing charge (APC). Authors will not be required to make a final decision about access to their article until it has been accepted. [Find out more about Transformative Journals](https://www.springernature.com/gp/open-research/transformative-journals)

Authors may need to take specific actions to achieve [compliance with funder and institutional open access mandates](https://www.springernature.com/gp/open-research/funding/policy-compliance-faqs). If your research is supported by a funder that requires immediate open access (e.g. according to [Plan S principles](https://www.springernature.com/gp/open-research/plan-s-compliance)) then you should select the gold OA route, and we will direct you to the compliant route where possible. For authors selecting the subscription publication route, the journal's standard licensing terms will need to be accepted, including [self-archiving-and-license-to-publish](https://www.nature.com/nature-portfolio/editorial-policies/self-archiving-and-license-to-publish). Those licensing terms will supersede any other terms that the author or any third party may assert apply to any version of the manuscript.

[REDACTED]

[REDACTED]

Reviewer #1:

None

Author Rebuttal, second revision:

This reviewer finds that the paper has improved in revision, but they would still like to see the issue of low sample size bias addressed more explicitly in the final text. Specifically, they strongly recommend the inclusion of a simulation study in which the overall bias in species-level estimates with sample size if evaluated. Please find attached a Script and Figure which they suggested were passed on to you where they show that the bias in species-level estimates is strongly associated with sample size, while community-level inference is not affected.

42At minimum, editorially we would like to see more explicit caution expressed in the text about making inferences on species with a small sample sizes. Once these additional changes are made we'll be happy in principle to publish it in Nature Ecology & Evolution, pending minor revisions to comply with our editorial and formatting guidelines.

We have reviewed the further feedback from Referee #1. Most of the inferences in our paper are focused on the fixed effects rather than the species random effects (which are the focus of his simulation).

We have made the following changes to the manuscript:

- 1. Add the range of populations per species included in our sample to the main text, where we reference the details in the supplementary table, for greater transparency in the main text (line 329 in tracked changes version).*
- 2. Add the following statement to the Supplementary results, at the first place where we discuss the species random effects (lines 382-384 in tracked changes version): "We acknowledge that sample sizes were uneven across species and families (Table S1), and that bias in estimates of these random effects is likely to be higher for species (and families) with fewer populations included in our sample."*
- 3. In our closing call for further research in the last paragraph of the main text (lines 521-522 in tracked changes version), we added the need for more sampling of underrepresented species to the previous emphasis on underrepresented regions.*

We trust that these changes suitably express the additional caution that you wished to see in the revision.

Final Decision Letter:

439th February 2024

Dear Dr Burton,

We are pleased to inform you that your Article entitled "Mammal responses to global changes in human activity vary by trophic group and landscape", has now been accepted for publication in *Nature Ecology & Evolution*.

Over the next few weeks, your paper will be copyedited to ensure that it conforms to *Nature Ecology and Evolution* style. Once your paper is typeset, you will receive an email with a link to choose the appropriate publishing options for your paper and our Author Services team will be in touch regarding any additional information that may be required

Due to the importance of these deadlines, we ask you please us know now whether you will be difficult to contact over the next month. If this is the case, we ask you provide us with the contact information (email, phone and fax) of someone who will be able to check the proofs on your behalf, and who will be available to address any last-minute problems . Once your paper has been scheduled for online publication, the Nature press office will be in touch to confirm the details.

Acceptance of your manuscript is conditional on all authors' agreement with our publication policies (see www.nature.com/authors/policies/index.html). In particular your manuscript must not be published elsewhere and there must be no announcement of the work to any media outlet until the publication date (the day on which it is uploaded onto our web site).

Please note that *Nature Ecology & Evolution* is a Transformative Journal (TJ). Authors may publish their research with us through the traditional subscription access route or make their paper immediately open access through payment of an article-processing charge (APC). Authors will not be required to make a final decision about access to their article until it has been accepted. Find out more about Transformative Journals

Authors may need to take specific actions to achieve compliance with funder and institutional open access mandates. If your research is supported by a funder that requires immediate open access (e.g. according to Plan S principles) then you should select the gold OA route, and we will direct you to the compliant route where possible. For authors selecting the subscription publication route, the journal's standard licensing terms will need to be accepted, including [a href="https://www.nature.com/nature-portfolio/editorial-policies/self-archiving-and-license-to-publish](https://www.nature.com/nature-portfolio/editorial-policies/self-archiving-and-license-to-publish). Those licensing terms will supersede any other terms that the author or any third party may assert apply to any version of the manuscript.

In approximately 10 business days you will receive an email with a link to choose the appropriate publishing options for your paper and our Author Services team will be in touch regarding any

44additional information that may be required.

We welcome the submission of potential cover material (including a short caption of around 40 words) related to your manuscript; suggestions should be sent to Nature Ecology & Evolution as electronic files (the image should be 300 dpi at 210 x 297 mm in either TIFF or JPEG format). Please note that such pictures should be selected more for their aesthetic appeal than for their scientific content, and that colour images work better than black and white or grayscale images. Please do not try to design a cover with the Nature Ecology & Evolution logo etc., and please do not submit composites of images related to your work. I am sure you will understand that we cannot make any promise as to whether any of your suggestions might be selected for the cover of the journal.

You can generate the link yourself when you receive your article DOI by entering it here: <http://authors.springernature.com/share>.

[REDACTED]

P.S. Click on the following link if you would like to recommend Nature Ecology & Evolution to your librarian <http://www.nature.com/subscriptions/recommend.html#forms>

** Visit the Springer Nature Editorial and Publishing website at www.springernature.com/editorial-and-publishing-jobs for more information about our career opportunities. If you have any questions please click here.**